# Beyond Description: Federated Adaptation via Semantic-Visual Prototype Alignment

**Jiarong Yang** [1]    **Yuan Liu** [1]

## Abstract

Adopting pre-trained Vision-Language Models (VLMs) in Federated Learning (FL) presents a promising avenue for mitigating data scarcity and heterogeneity. However, existing solutions suffer from high computational complexity or ineffective knowledge aggregation. To address these problems, we propose FedSPA (Federated Adaptation via Semantic-Visual Prototype Alignment). On the client side, FedSPA restricts local optimization to visual prototypes, enabling lightweight personalization. On the server side, we introduce a semantic alignment module that leverages client-uploaded prototypes to minimize a contrastive objective, aligning global semantic prototypes with heterogeneous visual distributions and thereby shifting the paradigm from traditional "learning-to-describe" (optimizing static prompts) to "learning-to-align". Extensive experiments demonstrate that FedSPA significantly outperforms state-of-the-art methods in both personalized and global benchmarks, while substantially reducing computational overhead. The code is available at https://github.com/eejiarong/FedSPA-main.

## 1. Introduction

Federated Learning (FL) (McMahan et al., 2017; Li et al., 2020) establishes itself as a prominent paradigm for collaborative machine learning, enabling distributed clients to train a global model without exposing private data. Despite its privacy benefits, FL faces a fundamental challenge: data heterogeneity (i.e., non-IID data) (Zhang et al., 2022a; Wang et al., 2024). The divergence between local distributions leads to "client drift", where the aggregated global model fails to generalize well to individual clients. To mitigate this, Personalized Federated Learning (PFL) is extensively studied, utilizing techniques such as regularization (Smith et al., 2017; Deng et al., 2020; T Dinh et al., 2020; Huang et al., 2021), meta-learning (Jiang et al., 2019; Fallah et al., 2020), and model decomposition (Collins et al., 2021; Liang et al., 2020; Oh et al., 2021) to tailor models for local clients. However, standard FL and PFL paradigms generally require training large vision backbones (e.g., ResNets, ViTs), leading to two critical bottlenecks: (i) high computational complexity: the compute-intensive back-propagation on massive parameters imposes a substantial computational burden on clients (Lee & Lee, 2021; Yang et al., 2023); (ii) data scarcity: training high-capacity models requires substantial labeled data, which is often scarce on individual clients (Guo et al., 2023).

To mitigate these limitations, large-scale Vision-Language Models (VLMs) (Jia et al., 2021; Kim et al., 2021) like CLIP (Radford et al., 2021) offer a paradigm shift. Instead of training from scratch, VLMs allow for parameter-efficient adaptation with frozen encoders to relieve computational burdens, while utilizing web-scale pre-trained knowledge to compensate for local data scarcity. Consequently, recent research shifts from training heavy backbones to adapting frozen VLMs, broadly categorized into three paradigms: (i) Federated prompt tuning (Guo et al., 2023; Luo et al., 2025) replaces static templates with learnable continuous vectors to steer the text encoder. This lightweight approach significantly reduces communication overhead by transmitting only prompt parameters rather than full model weights. (ii) Federated adapter-based adaptation (Ghiasvand et al., 2025) injects trainable modules (e.g., LoRA (Hu et al., 2022)) into intermediate layers of the frozen backbone. This enables deep feature modulation to capture task-specific knowledge without the prohibitive cost of full-model fine-tuning. (iii) Federated cache-based adaptation (Yi et al., 2025; Liu & Huang, 2025) prioritizes computational efficiency by constructing learnable key–value stores from visual features. By restricting optimization to this shallow feature space, these methods avoid back-propagation through the deep backbone, enabling a highly efficient training process.

Despite these advantages, these approaches suffer from high

[1]School of Electronic and Information Engineering, South China University of Technology, Guangzhou 510641, China. Correspondence to: Yuan Liu <eeyliu@scut.edu.cn>.

*Proceedings of the 43rd International Conference on Machine Learning*, Seoul, South Korea. PMLR 306, 2026. Copyright 2026 by the author(s).

computational complexity or ineffective knowledge aggregation. On the one hand, both prompt tuning and adapter-based methods incur high computational overheads, as they necessitate gradient back-propagation through the deep backbone. On the other hand, though cache-based methods gain efficiency via lightweight parameter optimization, they inherently suffer from severe data heterogeneity. Due to the diverse local data, the cached visual features exhibit large distributional shifts across clients. This divergence prevents traditional aggregation (e.g., FedAvg (McMahan et al., 2017)) from producing a coherent global model, since averaging parameters derived from disjoint feature spaces fails to capture a unified semantic representation.

To address these challenges, we propose FedSPA (Federated Adaptation via Semantic-Visual Prototype Alignment), which synergizes personalization and generalization by maintaining learnable personalized visual prototypes and global semantic prototypes. Specifically, on the client side, FedSPA facilitates efficient personalization by optimizing solely a lightweight, client-specific prototype matrix. This strategy enables decision boundaries to adapt to local distributions without incurring the excessive costs of deep gradient computation. On the server side, we introduce a semantic alignment module. Recognizing that static text descriptions fail to capture client diversity and client-side prompt tuning is computationally expensive, we propose to maintain and optimize a set of global semantic prototypes. By leveraging client-uploaded personalized visual prototypes as visual references, the server utilizes a contrastive objective to dynamically refine the global semantic prototypes. This process enforces robust alignment between global semantics and heterogeneous client visual distributions, shifting the paradigm from traditional "learning-to-describe" (optimizing static prompts) to "learning-to-align". These optimized semantic prototypes function as a robust global classifier. For personalized adaptation, they are integrated with local visual prototypes via a residual connection, yielding a composite classifier that synergizes global semantic knowledge with local visual nuances. Therefore, FedSPA achieves both personalized adaptation and global generalization. We summarize our contributions as follows:

- We propose FedSPA, which enables efficient personalization via lightweight client-side visual prototypes and facilitates global knowledge aggregation through learnable server-side semantic prototypes.

- We propose a semantic alignment module to bridge the semantic-visual gap, which establishes a robust alignment between global semantic knowledge and heterogeneous local visual distributions.

- Extensive experiments demonstrate that FedSPA achieves over $10\times$ computational efficiency improvement while outperforming state-of-the-art methods in both personalized and global benchmarks.

## 2. Related Work

### 2.1. Personalized Federated Learning

FL enables collaborative model training across a central server and multiple clients without sharing raw data. However, pronounced data and system heterogeneity often render a single global model inadequate. PFL extends FL to accommodate client heterogeneity, tailoring model parameters to individual clients while leveraging collaborative knowledge. Existing approaches generally fall into four paradigms: regularization-based methods that constrain local deviation from a global reference (Smith et al., 2017; Deng et al., 2020; T Dinh et al., 2020; Huang et al., 2021); meta-learning frameworks optimizing initialization for rapid local adaptation (Jiang et al., 2019; Fallah et al., 2020); architecture-based strategies that decouple shared and private components (Collins et al., 2021; Liang et al., 2020; Oh et al., 2021; Ma et al., 2022), with recent advances integrating knowledge distillation (Xie et al., 2024); and group-aware methods that cluster clients with similar distributions (Ghosh et al., 2020; Li et al., 2023; Lu et al., 2023a). However, traditional PFL typically involves training deep vision backbones from scratch. This imposes high computational complexity and labeled data requirements, motivating a shift toward pre-trained VLMs that offer robust zero-shot generalization.

### 2.2. Federated Learning with VLMs

VLMs such as CLIP (Radford et al., 2021) are trained at web scale on paired image–text data to learn aligned, language-grounded embeddings. By leveraging these strong semantic priors to enable data-efficient adaptation, VLMs minimize reliance on extensive local datasets (Jia et al., 2021; Kim et al., 2021). This makes them attractive foundations for FL, particularly in scenarios dominated by data scarcity and heterogeneity. By freezing the pre-trained backbones, existing VLM-based FL approaches harness the rich knowledge inherent in the encoders and generally fall into three categories: federated prompt tuning, federated adapter-based adaptation, and federated cache-based adaptation.

**Federated Prompt Tuning.** Federated prompt tuning replaces hand-crafted static templates with learnable continuous vectors to steer the frozen VLM (Zhou et al., 2022b;a). Pioneering works such as PromptFL (Guo et al., 2023) distribute the optimization of soft prompts to clients, significantly reducing communication bandwidth compared to full-model fine-tuning. Subsequent works focus on enhancing generalization across heterogeneous client-specific data distributions. Strategies include context-aware prompt

generation (Qiu et al., 2024), dual-prompt decoupling for domain knowledge (Zheng et al., 2025), and harmonized regularization to balance global consensus (Cui et al., 2024). To address data complexity, probabilistic modeling (Weng et al., 2024) and missing-modality alignment frameworks (Phung et al., 2025) are proposed. Furthermore, pFedMoAP (Luo et al., 2025) utilizes a mixture-of-experts strategy to capture diverse client semantics. Despite their flexibility, these methods necessitate gradient propagation through the text encoder, incurring excessive computational overhead on clients.

**Federated Adapter-Based Adaptation.** To adapt the image encoder of VLMs to downstream tasks, these methods inject lightweight trainable modules (e.g., LoRA (Hu et al., 2022)) into the frozen image encoder. FedCLIP (Lu et al., 2023b) utilizes attention-based adapters for feature alignment, while TriplePlay (Imteaj et al., 2024) employs LoRA to handle non-IID data efficiently. Recently, pFedMMA (Ghiasvand et al., 2025) introduces structured adapters with modality-specific projections. However, the requirement to back-propagate gradients into deep model layers prevents treating the backbone as a fully frozen black box, limiting efficiency on clients.

**Federated Cache-Based Adaptation.** Prioritizing maximum computational efficiency, cache-based methods function by constructing a key-value store of visual features (Support Set) to perform retrieval-based classification. This design allows for "training-free" or "shallow-training" adaptation (Zhang et al., 2022b; 2024a). CacheFL (Yi et al., 2025) initializes a class-balanced cache via generative synthesis and aggregates the updates of the lightweight cache model. FedPGA (Liu & Huang, 2025) further refines this by using prototype-guided caching to mitigate inter-client feature misalignment. Despite their efficiency, these methods inherently struggle with data heterogeneity. Acting as terminal classifiers, local caches are highly sensitive to distribution shifts, producing divergent models that hinder effective global aggregation.

To address the limitations of existing methods, we propose FedSPA, a framework that synergizes personalization and generalization by maintaining learnable personalized visual prototypes and global semantic prototypes. Unlike existing methods that achieve personalization via decoupling at either the network-layer level (e.g., FedRep (Collins et al., 2021)) or the prompt space (e.g., FedPGP (Cui et al., 2024)), FedSPA introduces a novel cross-modal representation-space decoupling. We separate the learning spaces at the embedding level: clients limit their optimization to lightweight visual prototypes to effectively capture local data distributions, while the server globally aligns the semantic prototypes via the proposed semantic alignment module to preserve shared knowledge. This design treats

the VLM backbone as a strictly frozen black box, shifting the paradigm from computationally heavy "learning-to-describe" to highly efficient "learning-to-align".

## 3. Methodology

### 3.1. Preliminaries

**Federated Learning Setup.** We consider a federated learning system comprising $K$ clients, where the $k$-th client holds a private local dataset $\mathcal{D}_k = \{(\mathbf{x}, y)\} \sim \mathcal{P}_k$. Due to statistical data heterogeneity, the local distributions $\mathcal{P}_k$ vary significantly across clients (i.e., the non-IID setting where $\mathcal{P}_i \neq \mathcal{P}_j$). Our objective is to collaboratively perform $N$-way classification over the label space $\mathcal{Y} = \{1, \ldots, N\}$, aiming to simultaneously achieve high accuracy in personalization on local distributions and robust generalization across diverse clients.

**VLM Backbone.** We employ a pre-trained VLM as the foundation. It consists of an image encoder $f(\cdot)$ and a text encoder $h(\cdot)$, projecting visual and textual inputs into a shared embedding space of dimension $d$. Specifically, for an input image $\mathbf{x}$, the image encoder extracts a normalized visual feature $\mathbf{f} = f(\mathbf{x}) \in \mathbb{R}^d$.

**Global Semantic Prototypes.** We maintain a set of learnable global semantic prototypes $\mathbf{Z} \in \mathbb{R}^{N \times d}$ on the server to represent the shared knowledge. To leverage the semantic priors of the pre-trained VLM, these prototypes are initialized by projecting fixed, hand-crafted prompts through the text encoder. Formally, for each class $n$ with class name $y_n$, the initialization is defined as:

$$\mathbf{z}_n^{(0)} = h(\mathcal{T}(y_n)), \tag{1}$$

where $\mathcal{T}(\cdot)$ denotes the prompt engineering function (e.g., $\mathcal{T}(y_n) = $ "a photo of a $y_n$").

### 3.2. Overview of the FedSPA Framework

In this subsection, we provide an overview of the FedSPA framework, which synergizes personalization and generalization via an alternating optimization strategy. The overall architecture is illustrated in Figure 1.

**Prediction Framework.** We establish two prediction schemes tailored for specific objectives. First, to achieve global generalization, we utilize the global semantic prototypes $\mathbf{Z}$ as a unified global classifier. Given a test image feature $\mathbf{f}$, the generalized prediction scores are defined as:

$$S_g(\mathbf{f}) = \mathbf{f}\mathbf{Z}^\top. \tag{2}$$

Second, to achieve local personalization, we model the client-side classification logits as the fusion of a local visual term and a global semantic term. Inspired by (Zhang et al.,

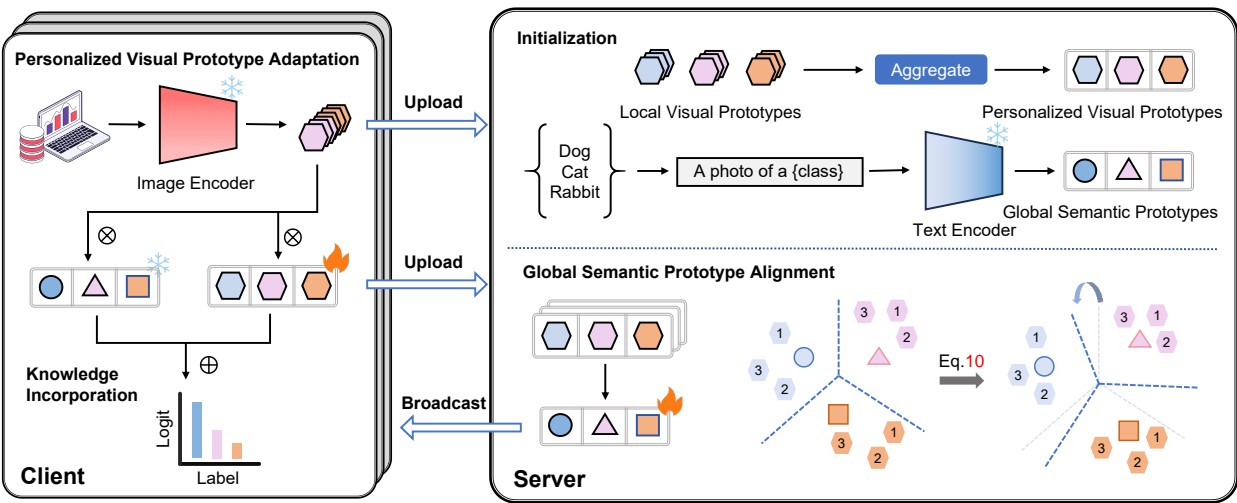

*Figure 1.* **Overview of the FedSPA framework.** Different shapes denote the personalized visual prototypes and the global semantic prototypes, while colors indicate different classes. Initialization: Each client constructs local prototypes and uploads them to the server; the server aggregates these local prototypes to initialize the personalized visual prototypes and, in parallel, initializes the global semantic prototypes from class names using a text encoder. Training: The server broadcasts the global semantic prototypes, which are kept frozen on each client. Clients then update their personalized visual prototypes using local features (see (7)) and upload the updated prototypes to the server. The server refines the global semantic prototypes by contrastively aligning them with the received visual prototypes (see (10)), thereby completing one communication round.

2022b), the personalized prediction scores are formulated as:

$$S_k(\mathbf{f}; \mathbf{V}_k, \mathbf{Z}) = \underbrace{\alpha\varphi(\mathbf{f}\mathbf{V}_k^\top)}_{\text{Local Visual Matching}} + \underbrace{\mathbf{f}\mathbf{Z}^\top}_{\text{Global Semantic Matching}},$$

(3)

where $\varphi(z) = \exp[-\beta(1-z)]$ with $\alpha, \beta > 0$. This formulation explicitly assigns distinct roles to two sets of decoupled parameters:

- $\mathbf{V}_k \in \mathbb{R}^{N \times d}$ (Personalized visual prototypes): A client-specific, learnable visual classifier. It is iteratively updated on private local data to capture distinct decision boundaries for personalization (detailed in Sec. 3.3).

- $\mathbf{Z} \in \mathbb{R}^{N \times d}$ (Global semantic prototypes): A globally shared set of learnable parameters maintained by the server. These prototypes are initialized using semantic priors and subsequently optimized to align with the visual distributions of all clients, promoting robust generalization (detailed in Sec. 3.4).

**Optimization Objective.** Our goal is to jointly optimize the global semantic prototypes $\mathbf{Z}$ and the personalized visual prototypes $\{\mathbf{V}_k\}_{k=1}^K$. We formulate this as a joint optimization problem:

$$\min_{\mathbf{Z}, \{\mathbf{V}_k\}_{k=1}^K} \sum_{k=1}^K \frac{|\mathcal{D}_k|}{|\mathcal{D}|} \mathcal{L}_k(\mathbf{V}_k, \mathbf{Z}),$$

(4)

where $\mathcal{L}_k$ denotes the empirical risk on client $k$. FedSPA employs an alternating optimization strategy: clients perform gradient-based updates on $\mathbf{V}_k$ over their private local data, while the server optimizes $\mathbf{Z}$ via contrastive learning utilizing the collected visual prototypes.

### 3.3. Client-Side: Personalized Visual Prototype Adaptation

**Initialization.** At round $t = 0$, each client $k$ computes the class-wise mean features using its local dataset $\mathcal{D}_k$:

$$\widehat{\mathbf{v}}_{k,n} = \frac{1}{|\mathcal{D}_{k,n}|} \sum_{(\mathbf{x},y) \in \mathcal{D}_{k,n}} f(\mathbf{x}),$$

(5)

where $\mathcal{D}_{k,n}$ denotes the set of samples for class $n$ on client $k$. Clients upload these local centroids to the server. The server then aggregates them to form the global visual prototypes $\overline{\mathbf{V}} = [\overline{\mathbf{v}}_1, \ldots, \overline{\mathbf{v}}_N]$, weighted by the local class frequencies:

$$\overline{\mathbf{v}}_n = \frac{\sum_k |\mathcal{D}_{k,n}| \cdot \widehat{\mathbf{v}}_{k,n}}{\sum_k |\mathcal{D}_{k,n}|}.$$

(6)

Finally, the server broadcasts $\overline{\mathbf{V}}$ to all clients. Each client initializes its personalized visual prototypes as $\mathbf{V}_k^{(0)} = \overline{\mathbf{V}}$, ensuring a consistent and robust starting point before personalization begins.

**Adaptation.** During subsequent training rounds ($t > 0$), with the global semantic prototypes $\mathbf{Z}$ (downloaded from the server) kept frozen, client $k$ optimizes the learnable visual

**Algorithm 1** Training Procedure of FedSPA

---

**Input:** Pre-trained VLM $\{f(\cdot), h(\cdot)\}$, Clients $\{1, \ldots, K\}$, Local Datasets $\mathcal{D}_k$.
**Output:** Personalized Visual Prototypes $\{\mathbf{V}_k\}_{k=1}^K$, Global Semantic Prototypes $\mathbf{Z}$.

**# Phase 1: Global Initialization ($t = 0$)**
**Server:** Initialize $\mathbf{Z}^{(0)}$ via text encoder $h(\cdot)$ using hand-crafted prompts.
**for** each client $k \in \{1, \ldots, K\}$ **in parallel do**
    Compute local class centroids $\widehat{\mathbf{v}}_{k,n}$ using $\mathcal{D}_k$; upload to server.
**end for**
**Server:** Aggregate global visual prototypes $\overline{\mathbf{V}}$; broadcast $\overline{\mathbf{V}}$ and $\mathbf{Z}^{(0)}$.
**Clients:** Initialize personalized visual prototypes $\mathbf{V}_k^{(0)} \leftarrow \overline{\mathbf{V}}$.

**# Phase 2: Alternating Optimization ($t = 1 \ldots T$)**
**for** round $t = 1 \ldots T$ **do**
    *// Client Step: Visual Adaptation (Personalization)*
    **for** each client $k \in \{1, \ldots, K\}$ **in parallel do**
        Receive global semantic prototypes $\mathbf{Z}^{(t-1)}$.
        **Freeze:** VLM backbone and $\mathbf{Z}^{(t-1)}$.
        **Update:** $\mathbf{V}_k^{(t)} \leftarrow \mathbf{V}_k^{(t-1)} - \eta \nabla_{\mathbf{V}_k} \mathcal{L}_k(\mathbf{V}_k, \mathbf{Z}^{(t-1)})$
        Upload updated prototypes $\mathbf{V}_k^{(t)}$ to server.
    **end for**

    *// Server Step: Semantic Alignment (Generalization)*
    Construct proxy dataset $\mathcal{V}^{(t)} = \{\mathbf{V}_k^{(t)}\}_{k=1}^K$.
    **Update:** $\mathbf{Z}^{(t)} \leftarrow \mathbf{Z}^{(t-1)} - \gamma \nabla_{\mathbf{Z}} \mathcal{L}_{\text{total}}(\mathbf{Z}; \mathcal{V}^{(t)})$
    Broadcast updated $\mathbf{Z}^{(t)}$ to clients.
**end for**

---

prototypes $\mathbf{V}_k$ via gradient descent to minimize the local objective defined in (4):

$$\mathbf{V}_k \leftarrow \mathbf{V}_k - \eta \cdot \nabla_{\mathbf{V}_k} \mathcal{L}_k(\mathbf{V}_k, \mathbf{Z}), \quad (7)$$

where $\eta$ is the learning rate. By strictly freezing the VLM backbone and only updating the low-dimensional matrix $\mathbf{V}_k$, clients efficiently adapt their decision boundaries to the local distribution $\mathcal{P}_k$ without heavy back-propagation costs.

### 3.4. Server-Side: Global Semantic Prototype Alignment

While personalized visual prototypes $\mathbf{V}_k$ capture local variations, the global semantic prototypes must remain aligned with these evolving visual representations to ensure generalization. We address this by optimizing the global semantic prototypes $\mathbf{Z}$ using a contrastive InfoNCE loss (Oord et al., 2018). Specifically, the server collects the updated local visual prototypes $\{\mathbf{V}_k\}_{k=1}^K$ from all clients to form a proxy dataset $\mathcal{V}$. We treat the global semantic prototype $\mathbf{z}_n$ (for

class $n$) as the anchor, and the corresponding client visual prototypes $\{\mathbf{v}_{k,n}\}_{k=1}^K$ as positive keys. To learn discriminative semantic prototypes, we minimize the following contrastive loss:

$$\mathcal{L}_{\text{InfoNCE}}(\mathbf{Z}; \mathcal{V})$$
$$= \sum_{n=1}^N \sum_{k=1}^K -\log \frac{\exp(\text{sim}(\mathbf{z}_n, \mathbf{v}_{k,n})/\tau)}{\sum_{j=1}^N \exp(\text{sim}(\mathbf{z}_n, \mathbf{v}_{k,j})/\tau)}, \quad (8)$$

where $\mathbf{v}_{k,n}$ denotes the $n$-th row of $\mathbf{V}_k$, $\text{sim}(\cdot, \cdot)$ represents cosine similarity, and $\tau$ is a temperature parameter. This objective explicitly forces the global semantic prototype $\mathbf{z}_n$ to be close to the visual prototypes of the same class $n$ from various clients, while pushing it away from visual prototypes of different classes $j \neq n$. Through this optimization, the server effectively aggregates the visual knowledge dispersed across clients into a compact, semantically aligned global representation.

Unconstrained optimization of (8) on heterogeneous personalized visual prototypes may lead to semantic drift, where the learned prototypes deviate excessively from the original semantic space of the pre-trained VLM. To mitigate this and preserve the generic knowledge embedded in the text encoder, we introduce a stability regularizer that maximizes the cosine similarity between the current prototype $\mathbf{z}_n$ and its initial state $\mathbf{z}_n^{(0)}$ (defined in (1)):

$$\mathcal{L}_{\text{reg}}(\mathbf{Z}) = -\frac{1}{N} \sum_{n=1}^N \text{sim}(\mathbf{z}_n, \mathbf{z}_n^{(0)}). \quad (9)$$

In summary, the server performs the update by minimizing the total composite loss:

$$\min_{\mathbf{Z}} \mathcal{L}_{\text{total}}(\mathbf{Z}) = \mathcal{L}_{\text{InfoNCE}}(\mathbf{Z}; \mathcal{V}) + \lambda \mathcal{L}_{\text{reg}}(\mathbf{Z}), \quad (10)$$

where $\lambda \geq 0$ is a hyperparameter controlling the strength of the regularization. Through this regularized optimization, the server effectively aggregates the dispersed visual knowledge into a compact global representation while maintaining semantic consistency with the pre-trained backbone.

### 3.5. Algorithm Summary

The complete training procedure of FedSPA is outlined in Algorithm 1. The framework orchestrates a collaborative learning process via an alternating optimization strategy:

- **Client-Side Visual Adaptation:** Clients download the global semantic prototypes $\mathbf{Z}$. They freeze these global semantics and the VLM backbone, performing gradient-based updates solely on their personalized visual prototypes $\mathbf{V}_k$ via (7). This adapts decision boundaries to local heterogeneous data.

- **Server-Side Semantic Alignment:** The server collects the personalized visual prototypes to form a proxy dataset $\mathcal{V}$. It then updates the global semantic prototypes $\mathbf{Z}$ by minimizing the regularized contrastive loss as (10). This step ensures that the global semantic prototypes are continuously refined to align with the evolving visual representations of the clients.

## 4. Experiments

### 4.1. Experimental Settings

**Implementation Details.** Unless otherwise stated, we simulate a federated learning system with $K = 10$ clients. Our framework is built upon the CLIP model, utilizing ResNet-50 (He et al., 2016) and ViT-B/16 (Dosovitskiy et al., 2020) as backbones, initialized with official pre-trained weights (Radford et al., 2021). The federated training spans $I = 10$ global communication rounds. Within each round, clients update their personalized visual prototypes $\mathbf{V}_k$ for $T = 5$ local epochs, while the server optimizes the global semantic prototypes $\mathbf{Z}$ for $T = 100$ epochs using the proxy dataset. Consistent with the few-shot training strategy, we employ the AdamW optimizer with a mini-batch size set equal to the number of shots for both optimization processes. Hyperparameters $\alpha$ and $\beta$, along with the learning rates, are configured following (Zhang et al., 2024a). To simulate statistical heterogeneity (Non-IID), we partition the data using a Dirichlet distribution $\mathrm{Dir}(\rho)$ over label ratios. Specifically, for each client, the proportion of samples for each class is drawn from $\mathrm{Dir}(\rho)$, creating imbalanced local class distributions. A smaller $\rho$ indicates a higher degree of data heterogeneity.

**Baselines.** We evaluate the proposed framework under two testing protocols to assess both generalization and personalization capabilities:

- **Generalization (FedSPA-G):** Inference is performed using the server-optimized global semantic prototypes $\mathbf{Z}$. We compare against federated cache-based methods, including FedPGA (Liu & Huang, 2025) and CacheFL (Yi et al., 2025).

- **Personalization (FedSPA-P):** Inference follows the residual fusion mechanism, where final predictions are generated by combining similarity scores from the client-specific personalized visual prototypes $\mathbf{V}_k$ with those from the global semantic prototypes $\mathbf{Z}$. We benchmark against three categories of baselines: (i) distributed variants of few-shot vision-language adaptation methods, specifically Tip-Adapter (Zhang et al., 2022b) and ECALP (Li et al., 2025), both of which personalize on local data; (ii) federated prompt tuning methods, including PromptFL (Guo et al., 2023) and

pFedMoAP (Luo et al., 2025); and (iii) the federated adapter-based method pFedMMA (Ghiasvand et al., 2025).

For a fair comparison, all baselines are re-implemented using identical CLIP backbones and subjected to consistent federated data partitions.

**Datasets.** We conduct extensive evaluations on a suite of 10 diverse image classification benchmarks: Flowers102 (Nilsback & Zisserman, 2008), Caltech101 (Fei-Fei et al., 2004), OxfordPets (Parkhi et al., 2012), StanfordCars (Krause et al., 2013), UCF101 (Soomro et al., 2012), FGVCAircraft (Maji et al., 2013), Food101 (Bossard et al., 2014), SUN397 (Xiao et al., 2010), DTD (Cimpoi et al., 2014), and EuroSAT (Helber et al., 2019). We adopt a few-shot setting, selecting 8 training samples per class, and evaluate on the full test sets; client-specific training and test subsets are partitioned via $\mathrm{Dir}(\rho)$.

### 4.2. Performance Evaluation

**Performance Comparison under Extreme Heterogeneity.** We first evaluate the performance of FedSPA under an extreme non-IID setting ($\rho = 0.1$) with 8 shots per class. The experiments are conducted using both ResNet-50 and ViT-B/16 backbones across ten datasets. To ensure sufficient personalization, we extend local training to 100 iterations in the final round ($I = 10$) for FedSPA. For fairness, lightweight cache-based methods—namely FedPGA, CacheFL, and Tip-Adapter—are evaluated under the same total local iteration budget. The results, as detailed in Table 1, demonstrate that FedSPA achieves state-of-the-art performance in both global generalization and local personalization metrics. Specifically, for global generalization, FedSPA-G consistently outperforms baselines across different backbones, achieving mean accuracies of $69.59\%$ (ResNet-50) and $75.89\%$ (ViT-B/16), significantly surpassing the second-best methods ($63.96\%$ and $70.33\%$). The advantage is most distinct on fine-grained datasets like Flowers102, validating that our aligned semantic prototypes capture more robust representations than the simple aggregation of visual prototypes (e.g., via FedAvg) used in CacheFL and FedPGA. Regarding personalization, FedSPA-P demonstrates superior adaptability, attaining $79.89\%$ (ResNet-50) and $84.91\%$ (ViT-B/16), consistently outperforming the leading federated prompt tuning baseline pFedMoAP ($76.70\%$ and $82.56\%$). Notably, it exceeds the local-only Tip-Adapter by approximately $6\%$ on both backbones. This confirms that the global knowledge embedded in our aligned semantic prototypes is beneficial for personalization, effectively mitigating overfitting in heterogeneous few-shot scenarios.

**Performance Comparison under Varying Shots per Class.** Consistent with the previous configuration, we in-

*Table 1.* **Test accuracy (%) on various datasets with different backbones.** We compare **FedSPA** against state-of-the-art baselines. **Bold** indicates the best performance within each group (Generalization or Personalization). Gray rows indicate our proposed method.

| Backbone | Method | Flower | DTD | Pets | Cars | UCF | Caltech | Food | SUN | Aircraft | EuroSAT | Mean |
|---|---|---|---|---|---|---|---|---|---|---|---|---|
| | CLIP-RN50 | 65.94 | 42.20 | 85.80 | 55.66 | 61.56 | 85.80 | 77.30 | 58.53 | 17.19 | 37.59 | 58.76 |
| | *Generalization* | | | | | | | | | | | |
| | FedPGA | 71.21 | 49.88 | 86.40 | 60.86 | 65.79 | 89.05 | **78.22** | 64.80 | 19.56 | 53.84 | 63.96 |
| | CacheFL | 67.07 | 45.27 | 86.24 | 57.01 | 64.84 | 89.82 | 78.05 | 62.31 | 18.00 | 52.96 | 62.15 |
| ResNet-50 | **FedSPA-G** | **91.23** | **58.10** | **88.01** | **64.28** | **72.40** | **90.67** | 73.54 | **66.89** | **21.54** | **69.19** | **69.59** |
| | *Personalization* | | | | | | | | | | | |
| | Tip-Adapter | 86.22 | 63.40 | **91.83** | 76.30 | 78.38 | 84.92 | 73.51 | 72.30 | 44.55 | 61.52 | 73.29 |
| | ECALP | 66.40 | 59.68 | 70.70 | 58.44 | 63.40 | 67.85 | 62.09 | 60.77 | 41.35 | 49.46 | 60.01 |
| | PromptFL | 72.07 | 49.97 | 85.04 | 59.12 | 64.32 | 89.78 | 77.44 | 63.93 | 20.83 | 50.49 | 63.30 |
| | pFedMoAP | 87.43 | 73.44 | 91.28 | 73.10 | 78.21 | 92.92 | 80.77 | 72.28 | 43.11 | **74.45** | 76.70 |
| | **FedSPA-P** | **95.68** | **74.51** | 91.29 | **79.80** | **82.76** | **95.19** | **82.48** | **79.18** | **44.82** | 73.19 | **79.89** |
| | CLIP-ViTB/16 | 71.38 | 44.39 | 89.07 | 65.27 | 66.75 | 92.94 | 86.11 | 62.55 | 24.84 | 47.77 | 65.11 |
| | *Generalization* | | | | | | | | | | | |
| | FedPGA | 75.15 | 52.90 | 90.32 | 70.10 | 73.43 | 94.24 | **86.54** | 69.95 | 28.89 | 61.80 | 70.33 |
| | CacheFL | 72.35 | 48.82 | 89.94 | 67.16 | 72.59 | **94.48** | 86.45 | 67.37 | 25.98 | 60.80 | 68.59 |
| | **FedSPA-G** | **93.59** | **63.30** | **92.12** | **72.85** | **79.06** | 93.18 | 84.95 | **71.45** | **30.93** | **77.47** | **75.89** |
| ViT-B/16 | *Personalization* | | | | | | | | | | | |
| | Tip-Adapter | 88.96 | 65.26 | **94.75** | 83.39 | 85.19 | 85.86 | 80.08 | 74.04 | 53.08 | 71.31 | 78.19 |
| | ECALP | 68.63 | 61.17 | 73.38 | 65.64 | 65.98 | 69.27 | 68.32 | 63.89 | 51.35 | 55.15 | 64.27 |
| | PromptFL | 84.52 | 59.09 | 91.44 | 68.03 | 74.56 | 93.92 | 84.59 | 69.87 | 29.14 | 67.32 | 72.24 |
| | pFedMoAP | 91.86 | 76.38 | 93.01 | 79.44 | 85.33 | 96.83 | 88.28 | 77.13 | 55.02 | **82.35** | 82.56 |
| | pFedMMA | 78.24 | 67.69 | 94.33 | 68.97 | 78.07 | 97.08 | **89.71** | 71.96 | 34.08 | 81.62 | 76.17 |
| | **FedSPA-P** | **97.53** | **76.65** | 94.13 | **85.30** | **88.01** | **97.69** | 88.72 | **82.98** | **56.87** | 81.20 | **84.91** |

vestigate the sensitivity of FedSPA to the number of shots per class. We evaluate both generalization (FedSPA-G) and personalization (FedSPA-P) performance on the ResNet-50 backbone across all ten datasets. We vary the number of shots per class $k \in \{1, 2, 4, 8, 16\}$ while maintaining the heterogeneity parameter at $\rho = 0.1$. The results, illustrated in Figure 2, reveal a clear positive correlation between accuracy and the number of shots. Notably, FedSPA demonstrates remarkable robustness in the low-data regime (e.g., 2-shot and 4-shot settings). While the baseline method pFedMoAP performs poorly with scarce data, FedSPA-P maintains high performance even with minimal local samples, ranking second only to Tip-Adapter. In higher-shot settings, although pFedMoAP exhibits significant performance gains and surpasses other baselines, FedSPA-P continues to slightly outperform it, demonstrating consistent superiority. This suggests that our server-side alignment of global semantic prototypes effectively compensates for the scarcity of local visual samples.

### 4.3. Ablation Study

**Impact of Global Semantic Prototype Alignment.** To validate the effectiveness of our server-side strategy in aligning

*Table 2.* **Ablation study on the effectiveness of semantic alignment** ($\rho = 0.1$)**.** We compare FedSPA against Local-Only and FedAvg-based baselines. The "**Mean Acc.**" column represents the average test accuracy calculated across all 10 datasets.

| Method | Generalization | | Personalization | | |
|---|---|---|---|---|---|
| | FedAvg | **FedSPA-G** | FedAvg | Local-Only | **FedSPA-P** |
| **Mean Acc.** | 59.18 | **69.68** | 64.55 | 72.82 | **77.47** |

global semantics with local prototypes, we introduce two ablated variants: (i) a Local-Only variant lacking global guidance, and (ii) a FedAvg-based variant using naive prototype aggregation. First, the Local-Only Adaptation (w/o Alignment) baseline establishes a performance lower bound in the absence of global guidance. In this setting, clients optimize their personalized visual prototypes $\mathbf{V}_k$ exclusively on private data. Second, we implement a FedAvg-based Alignment baseline, where the server updates global semantic prototypes via naive averaging ($\mathbf{Z}' \leftarrow \frac{1}{K} \sum_k \mathbf{V}_k$). For a fair comparison, this variant employs the identical residual fusion mechanism during inference for personalization and utilizes the averaged $\mathbf{Z}'$ for generalization. The results in Table 2 underscore the limitations of naive parameter averaging (FedAvg) under data heterogeneity. Specifically, severe

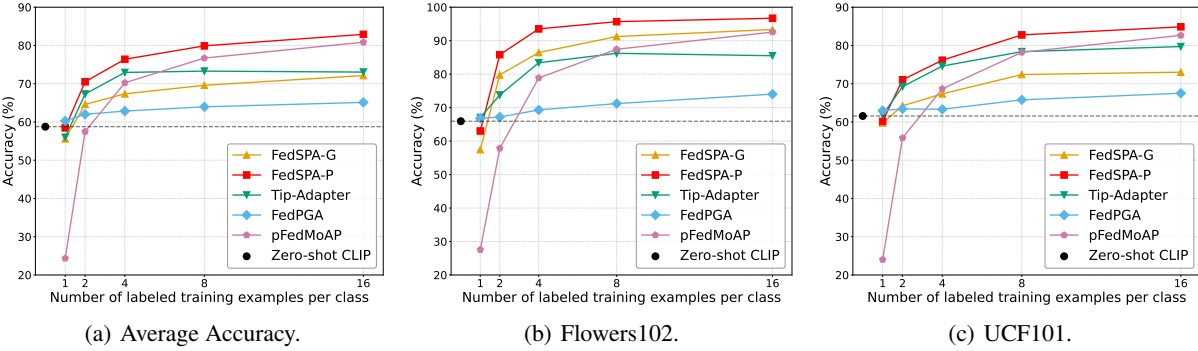

(a) Average Accuracy.     (b) Flowers102.     (c) UCF101.

*Figure 2.* **Performance comparison under varying shots per class.** We evaluate FedSPA and baselines using the ResNet-50 backbone. (a) Average test accuracy calculated across all 10 datasets.

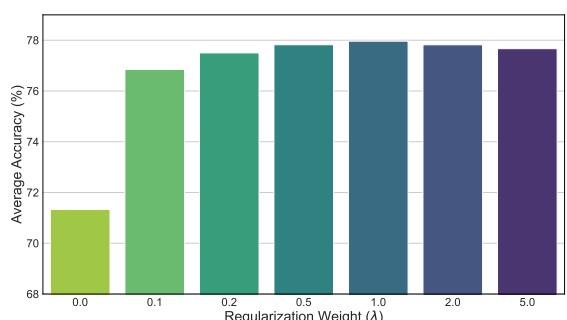

*Figure 3.* **Impact of regularization weight** $\lambda$**.** We evaluate the sensitivity of the proposed method to $\lambda$ across all ten datasets.

*Table 3.* **Average efficiency comparison across ten datasets**. We report the average local execution time (in seconds) and accuracy. **FedSPA-P (Ext.)** extends local training to 100 iterations in the final round for enhanced personalization. The **"Speedup"** column illustrates the relative time efficiency compared to FedAvg.

| Method | Time (s) ↓ | Acc. (%) ↑ | Speedup (×) ↑ |
|---|---|---|---|
| FedAvg | 59.98 | - | 1.0× |
| pFedMMA | 96.36 | 76.17 | 0.6× |
| pFedMoAP | 83.60 | 82.56 | 0.7× |
| **FedSPA-P** | **1.35** | 83.02 | **44.4×** |
| **FedSPA-P (Ext.)** | 3.60 | **84.91** | 16.7× |

heterogeneity induces client model divergence, hindering effective global aggregation. Consequently, FedAvg fails to construct a robust global representation, yielding inferior generalization accuracy ($59.18\%$) compared to FedSPA-G ($69.68\%$). More critically, in the personalization setting, FedAvg ($64.55\%$) underperforms even the Local-Only lower bound ($72.82\%$), suggesting that naive aggregation introduces interference rather than gain. In contrast, FedSPA-P ($77.47\%$) effectively mitigates these issues, validating the effectiveness of the proposed semantic alignment module.

**Sensitivity to Regularization Strength** $\lambda$**.** We further investigate the sensitivity of FedSPA to the regularization coefficient $\lambda$ in (10), which governs the trade-off between adapting to heterogeneous visual distributions and preserving pre-trained semantic priors. Specifically, we evaluate model performance by varying $\lambda$ across the set $\{0, 0.1, 0.2, 0.5, 1.0, 2.0, 5.0\}$ under the 8-shot setting with fixed heterogeneity ($\rho = 0.1$), using a ResNet-50 backbone. The results, illustrated in Figure 3, indicate that setting $\lambda = 0$ (unconstrained optimization) risks semantic drift, where updated prototypes deviate excessively from the robust VLM embedding space, leading to a significant drop in performance. Conversely, an excessively large $\lambda$ enforces rigid adherence to the initialization, stifling the necessary adaptation to client-specific variations and causing a slight

performance decline. Consequently, maintaining $\lambda$ within an intermediate range ($0.5 \sim 2.0$) yields consistently high performance.

### 4.4. Efficiency Comparison

To assess the computational efficiency of deployment on clients, we benchmark FedSPA against FedAvg, pFedMMA, and pFedMoAP. We report the average local execution time across ten datasets using the ViT-B/16 backbone, where FedAvg serves solely as a baseline for execution time. With the exception of FedSPA-P (Ext.), which extends local training to 100 iterations in the final round, all methods are configured with 5 local iterations and 10 global rounds. As shown in Table 3, FedSPA exhibits superior computational efficiency, achieving a $44.4\times$ speedup over FedAvg. Notably, this acceleration does not come at the expense of performance. While pFedMoAP requires $83.60$s to reach $82.56\%$ accuracy, FedSPA-P attains superior accuracy ($83.02\%$) in just $1.35$s. Furthermore, FedSPA-P (Ext.) yields the highest accuracy of $84.91\%$ while maintaining a $16.7\times$ speed advantage over the baseline.

### 4.5. Privacy Analysis

FedSPA can be extended with feature-level local differential privacy (LDP) by adding Gaussian noise before local

*Table 4.* **Utility of FedSPA under feature-level DP noise.** We report the test accuracy under different Gaussian noise levels $\sigma$ and local training epochs $T$. The $\epsilon_{wc}$ column denotes the conservative worst-case privacy budget.

| Regime | $\sigma$ | $\epsilon_{wc}$ | Accuracy (%) | | |
|---|---|---|---|---|---|
| | | | $T = 5$ | $T = 10$ | $T = 20$ |
| No noise | 0 | $\infty$ | 79.89 | – | – |
| Small | 0.10 | 96.8 | 78.80 | 78.99 | 79.00 |
| Medium | 0.24 | 40.3 | 76.92 | 77.35 | 77.73 |
| Large | 0.48 | 20.2 | 72.96 | 73.86 | 74.58 |

prototype construction. Since the frozen VLM produces $L_2$-normalized features, replacing one input sample has sensitivity at most $\Delta \leq 2$. Each client perturbs its local feature as

$$\tilde{f}(\mathbf{x}) = f(\mathbf{x}) + \mathcal{N}(0, \sigma^2\mathbf{I}), \quad (11)$$

which provides an $(\epsilon, \delta)$-LDP guarantee under the Gaussian mechanism. By the post-processing property of differential privacy, prototype averaging and server-side aggregation preserve this guarantee. Table 4 reports the utility of FedSPA under different DP noise levels, where the worst-case privacy budget is approximated as $\epsilon_{wc} \approx 9.68/\sigma$ with $\delta = 10^{-5}$. FedSPA remains robust under small and medium perturbations, incurring only a $1.09\%$ accuracy drop at $T = 5$ when $\sigma = 0.10$. Increasing the number of local epochs further reduces the utility degradation caused by noise injection, achieving $74.58\%$ accuracy even under the large-noise regime. We note that $\epsilon_{wc}$ is a conservative per-feature worst-case bound. In practice, the server typically observes noisy class-wise prototypes aggregated from multiple local features rather than isolated instance features. This aggregation provides an additional empirical privacy benefit by making feature inversion from a single noisy prototype substantially more ill-posed.

## 5. Conclusion

We propose FedSPA, a framework that harmonizes local visual personalization with global semantic alignment via an alternating optimization strategy. By constraining local optimization to lightweight client-side visual prototypes and shifting the paradigm from prompt tuning to dynamic semantic alignment, FedSPA addresses data heterogeneity while achieving highly efficient local computation. Our experiments demonstrate that FedSPA outperforms state-of-the-art methods in accuracy with a computational efficiency gain of over $10\times$, offering a highly efficient pathway for deploying VLMs in federated settings.

**Limitations.** FedSPA requires semantically meaningful labels to initialize global semantic prototypes via the text encoder. In domains where labels are merely anonymized IDs

(e.g., class_1) without textual descriptions, the framework cannot fully exploit the VLM's language priors. Furthermore, server computation scales linearly with the number of classes and clients ($N$ and $K$). Developing efficient contrastive alignment via class/client sampling for ultra-large-scale datasets is left for future work.

## Acknowledgements

This work was supported in part by the Nature and Science Funding of Guangdong Province under Grant 2026A1515011230 and in part by the Science and Technology Program of Shenzhen under Grant ZDCY20250901112705007.

## Impact Statement

FedSPA significantly lowers the computational and communication barriers to the democratization of VLMs in federated edge networks. By eliminating deep-backbone back-propagation on edge clients, it contributes to "Green AI" by drastically reducing carbon footprints. Ensuring ethical use and robust privacy measures remains a shared responsibility in decentralized ML.

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

# A. Detailed Experimental Settings

In this section, we provide a comprehensive overview of the implementation details, hyperparameters, and computing platform used in our experiments.

*Table 5.* **General experimental environment and protocols.** These settings are consistent across all comparative baselines unless otherwise noted.

| Configuration Item | Value / Specification |
|---|---|
| *Hardware & Platform* | |
| GPU | NVIDIA GeForce RTX 4080 |
| Deep Learning Framework | PyTorch |
| Random Seed | 1 |
| *Model Architecture* | |
| Backbones | ResNet-50, ViT-B/16 (CLIP Visual Encoder) |
| Initialization | Official OpenAI pre-trained weights (Radford et al., 2021) |
| *Federated Learning Setup* | |
| Total Clients ($K$) | 10 |
| Data Partition (Non-IID) | Dirichlet Distribution with $\rho = 0.1$ |
| *Optimization* | |
| Optimizer | AdamW |
| Batch Size | Equal to $k$-shot per class (e.g., 1, 2, 4, 8, 16) |

*Table 6.* **Detailed hyperparameter configurations and iteration settings.** This table details the global communication rounds $I$, local iterations $T$, and specific hyperparameter sources for each method. The **"Applicable Exp."** column indicates the specific Tables or Figures where these settings are applied. "–" indicates the parameter is not applicable.

| Method | $I$ | $T$ | Applicable Exp. | Other Hyperparameters |
|---|---|---|---|---|
| **FedSPA (Ours)** | 10 | 5 | Tab. 2, Tab. 3, Fig. 3 | Server Iterations set to 100 for (10). Parameters $\alpha, \beta, \lambda$ and Learning Rates follow (Zhang et al., 2024a). |
| | 10 | 100* | Tab. 1, Tab. 3, Fig. 2 | |
| **FedPGA** | – | 150 | Tab. 1, Fig. 2 | Follows (Zhang et al., 2024a). |
| **Tip-Adapter** | – | 150 | Tab. 1, Fig. 2 | Follows (Zhang et al., 2024a). |
| **CacheFL** | 30 | 5 | Tab. 1 | Follows (Zhang et al., 2024a). |
| **ECALP** | – | – | Tab. 1 | Follows (Li et al., 2025). |
| **PromptFL** | 10 | 5 | Tab. 1 | Follows (Luo et al., 2025). |
| **pFedMoAP** | 10 | 5 | Tab. 1, Tab. 3, Fig. 2 | Follows (Luo et al., 2025). |
| **pFedMMA** | 10 | 5 | Tab. 1, Tab. 3 | Follows (Ghiasvand et al., 2025). |

\* For FedSPA (Ext.), $T = 100$ is applied only in the final global round; previous rounds use $T = 5$.

## A.1. General Federated Environment

Table 5 summarizes the common configurations used across all experiments, including the hardware platform, backbone architectures, and federated learning protocols.

## A.2. Method-Specific Hyperparameters

Table 6 provides a comprehensive overview of the global communication rounds $I$, local training iterations $T$, and specific hyperparameter sources for all comparative methods. To ensure a fair comparison, prompt-learning and adapter-based baselines (e.g., PromptFL, pFedMoAP, pFedMMA) are implemented strictly following the optimal settings reported in their original papers. Cache-based methods (e.g., FedPGA, Tip-Adapter, CacheFL) generally follow the configurations outlined in (Zhang et al., 2024a). For our proposed FedSPA, we specify the server-side optimization iterations and adopt the hyperparameter configuration—including $\alpha, \beta, \lambda$ and learning rates—from (Zhang et al., 2024a).

# B. Additional Privacy Discussion

This section complements the privacy analysis in Section 4.5 by further discussing label leakage risks in semantic initialization and the privacy advantages of FedSPA over existing FL paradigms.

## B.1. Mitigating Label Leakage Risks in Semantic Initialization

A potential security concern when deploying VLMs in federated networks is that server-side text encoding might reveal raw class names, posing label leakage risks. In the FedSPA framework, this risk can be entirely avoided through a decentralized initialization strategy. Since the text encoder is only used once during the "Global Initialization" phase, clients can locally compute these initial semantic vectors $\mathbf{z}_n^{(0)}$ using their local class names and upload only the resulting vectors. The server then aggregates these vectors to initialize $\mathbf{Z}^{(0)}$. Thus, the server never accesses raw text, completely eliminating label leakage risks.

## B.2. Privacy Advantages over Existing FL Paradigms

Compared to existing FL paradigms, FedSPA demonstrates superior robustness against privacy leakage. Specifically:

- **Vs. Gradient-based FL:** Traditional FL methods (McMahan et al., 2017) require transmitting high-dimensional model weights or gradients, rendering them vulnerable to severe privacy leakage via gradient inversion attacks. FedSPA strictly transmits low-dimensional, aggregated features.

- **Vs. Federated Prompt Tuning:** Methods such as PromptFL (Guo et al., 2023) and pFedMoAP (Luo et al., 2025) require sharing textual prompts, which inevitably exposes sensitive class names to the server. As detailed in Appendix B.1, FedSPA can compute text features strictly locally, avoiding label leakage entirely.

- **Vs. Cache/Prototype-based FL:** Compared to other prototype-based FL approaches (e.g., FedPGA (Liu & Huang, 2025)), FedSPA does not introduce additional privacy exposure, while naturally supporting formal LDP guarantees.

# C. Comprehensive System Overhead and Scalability Analysis

In this section, we evaluate the system overhead of the FedSPA framework from two perspectives: the client-side communication and computation workloads, and the server-side computational scalability.

## C.1. Client-Side Communication and Computational Overhead

Since FedSPA transmits $N \times d$-dimensional prototype matrices, its communication traffic scales linearly with the number of classes $N$. To evaluate this scalability, we quantify the exact communication overhead (total upload and download across 10 global rounds) and the local computational execution time per client. We benchmark FedSPA against a representative prompt-tuning method (e.g., pFedMoAP), which transmits a fixed number of soft prompts ($16 \times 512$ parameters) regardless of the dataset, resulting in a constant traffic of $0.63$ MB.

As shown in Table 7, while FedSPA's communication traffic dynamically scales with $N$, it remains extremely lightweight even under extreme scaling scenarios. For instance, on the fine-grained SUN397 dataset ($N = 397$), the absolute maximum traffic for the entire training process is only $15.51$ MB. This overhead is negligible under modern network bandwidths, especially when compared to transmitting full deep learning models (typically $> 300$ MB).

More importantly, in exchange for this marginal increase in communication, FedSPA achieves a massive $> 100\times$ compu-

*Table 7.* **Comparison of total communication traffic (10 global rounds) and local execution time on the client side.**

| Dataset (Classes $N$) | Method | Total Traffic (MB) | Local Exec. Time (s) |
|---|---|---|---|
| **EuroSAT** ($N = 10$) | pFedMoAP | 0.63 | 17.59 |
| | **FedSPA** | **0.39** | **0.21** |
| **Food101** ($N = 101$) | pFedMoAP | **0.63** | 40.06 |
| | **FedSPA** | 3.95 | **1.21** |
| **SUN397** ($N = 397$) | pFedMoAP | **0.63** | 441.40 |
| | **FedSPA** | 15.51 | **4.17** |

*Table 8.* **Ablation on server-side optimization epochs ($T_{server}$) regarding execution time (seconds per round) and test accuracy (%).** Experiments are conducted to observe the computational scaling with respect to class count $N$ and client count $K$.

| Dataset ($N, K$) | $T_{server} = 10$ | | | $T_{server} = 50$ | | | $T_{server} = 100$ (**Default**) | | |
|---|---|---|---|---|---|---|---|---|---|
| | Time (s) | Acc-G (%) | Acc-P (%) | Time (s) | Acc-G (%) | Acc-P (%) | Time (s) | Acc-G (%) | Acc-P (%) |
| **OxfordPets** (37, 10) | 0.99 | 87.63 | 90.63 | 4.61 | 87.87 | 91.04 | 9.29 | 88.01 | 91.29 |
| **UCF101** (101, 10) | 2.92 | 70.34 | 82.09 | 10.30 | 71.27 | 82.11 | 30.42 | 72.40 | 82.76 |
| **StanfordCars** (196, 20) | 9.95 | 64.95 | 78.32 | 49.54 | 63.87 | 78.74 | 99.28 | 63.10 | 78.55 |

tational speedup over pFedMoAP on the client side (e.g., 4.17s vs. 441.40s on SUN397). This efficiency makes FedSPA suitable for real-world edge devices with strictly constrained computing resources.

### C.2. Server-Side Computational Overhead

In our main experiments, we set the default server-side optimization epochs to $T_{server} = 100$, based on the standard FL premise that the central server possesses abundant computational resources. Crucially, optimizing the server-side InfoNCE loss in FedSPA strictly involves lightweight, parallelized matrix multiplications over low-dimensional semantic prototypes ($K \times N \times d$, where $d = 512$). Consequently, the empirical time complexity on the server scales linearly as $\mathcal{O}(T_{server} \cdot K \cdot N \cdot d)$.

To demonstrate this scalability, we measure the per-round wall-clock time and the corresponding generalization/personalization accuracy across varying $T_{server}$, $K$, and $N$ using an NVIDIA RTX 4080 GPU. As detailed in Table 8, the execution time scales smoothly and linearly with $T_{server}$. Remarkably, the model's accuracy remains highly robust, exhibiting negligible fluctuations even when the server optimization budget is drastically reduced. Therefore, in severely resource-constrained server scenarios, we can safely reduce $T_{server}$ to 10 or 50 to ensure extreme computational efficiency without sacrificing the final model performance.

## D. Analysis of Convergence and Client Drift

In this section, we analyze the convergence behavior of FedSPA from two perspectives: the stability of the model's accuracy over extended communication rounds, and the mitigation of feature divergence (client drift) in the localized prototype space.

### D.1. Convergence over Extended Communication Rounds

We extend the FL process from 10 to 15 global communication rounds under the extreme heterogeneity setting ($\rho = 0.1$). As demonstrated in Table 9, the results clearly indicate that FedSPA safely converges within 10 rounds, exhibiting stability without oscillation in subsequent rounds. Furthermore, we note that $I = 10$ communication rounds is a standard evaluation protocol widely adopted by recent SOTA few-shot FL frameworks (e.g., pFedMoAP (Luo et al., 2025), pFedMMA (Ghiasvand et al., 2025)).

### D.2. Quantitative Analysis of Prototype Divergence

To verify that the server-side semantic alignment effectively mitigates client drift and facilitates feature-space convergence under extreme data heterogeneity, we introduce the **Intra-Class Cross-Client Cosine Distance**. This metric calculates the

*Table 9.* **Average accuracy (%) across 10 datasets extended to 15 communication rounds ($\rho = 0.1$).**

| Global Round | 10 | 11 | 12 | 13 | 14 | 15 |
|---|---|---|---|---|---|---|
| **FedSPA-G Acc. (%)** | 69.59 | 69.62 | 69.57 | 69.29 | 69.55 | 69.56 |
| **FedSPA-P Acc. (%)** | 79.89 | 79.90 | 80.00 | 79.98 | 80.13 | 80.03 |

*Table 10.* **Intra-Class Cross-Client Cosine Distance across 10 datasets using the ResNet-50 backbone ($\rho = 0.1$).** Lower values indicate better cross-client representation alignment.

| Method | Flower | DTD | Pets | Cars | UCF | Caltech | Food | SUN | Aircraft | EuroSAT | Mean |
|---|---|---|---|---|---|---|---|---|---|---|---|
| **Local-Only** | 0.1120 | 0.3090 | 0.0086 | 0.0971 | 0.1729 | 0.1912 | 0.2745 | 0.2809 | 0.4102 | 0.0059 | 0.1862 |
| **FedSPA** | **0.0195** | **0.2702** | **0.0024** | **0.0418** | **0.1190** | **0.1014** | **0.0808** | **0.1625** | 0.4554 | **0.0013** | **0.1254** |

average pairwise divergence among personalized visual prototypes of the same class across different clients.

Formally, let $\mathcal{K}_n$ denote the set of clients possessing data for class $n$. We exclusively consider the subset of valid classes $\mathcal{N}_{valid}$ shared by at least two clients (i.e., $|\mathcal{K}_n| \geq 2$). Let $\mathbf{v}_{i,n}$ and $\mathbf{v}_{j,n}$ represent the $L_2$-normalized visual prototypes for class $n$ on clients $i$ and $j$, respectively. The overall divergence $\mathcal{D}$ is formulated as the mean of the pairwise distances across all valid classes:

$$\mathcal{D} = \frac{1}{|\mathcal{N}_{valid}|} \sum_{n \in \mathcal{N}_{valid}} \left( \frac{2}{|\mathcal{K}_n|(|\mathcal{K}_n| - 1)} \sum_{i \in \mathcal{K}_n} \sum_{\substack{j \in \mathcal{K}_n \\ j > i}} \left( 1 - \text{sim}(\mathbf{v}_{i,n}, \mathbf{v}_{j,n}) \right) \right), \tag{12}$$

where $\text{sim}(\cdot, \cdot)$ denotes the cosine similarity. Lower values of $\mathcal{D}$ indicate better cross-client representation alignment and reduced client drift. As illustrated in Table 10, under extreme heterogeneity ($\rho = 0.1$), naive local-only updates yield a mean cross-client distance of $0.1862$. In contrast, with the integration of FedSPA's server-side alignment module, the average cross-client distance is reduced to $0.1254$ (an approximate $32.7\%$ reduction). This empirical evidence demonstrates that our alternating semantic alignment mechanism effectively mitigates client drift by encouraging local feature spaces to converge toward a more consistent global representation.

*Table 11.* **Average accuracy (%) across 10 datasets comparing different prompt initialization strategies.**

| Metric | Prompt Initialization Strategy | Average Acc. (%) |
|---|---|---|
| **FedSPA-G** | Simple Template: `"a photo of a [class]"` | 69.59 |
| | LLM-Generated Rich Descriptions | **69.72** |
| **FedSPA-P** | Simple Template: `"a photo of a [class]"` | 79.89 |
| | LLM-Generated Rich Descriptions | **79.92** |

## E. Robustness to Semantic Prompt Initialization

To explore the sensitivity of FedSPA to prompt engineering during the Global Initialization phase, we replace the simple hand-crafted template (`"a photo of a [class]"`) with rich, detailed descriptions generated by a Large Language Model (e.g., GPT) (Zhang et al., 2024b; Li et al., 2025). These LLM-generated descriptions include comprehensive visual traits such as shape, texture, and context for each class. The averaged text features of these diverse descriptions are then utilized as the initial global semantic prototypes $\mathbf{z}_n^{(0)}$. The experimental results, averaged across all 10 datasets, are summarized in Table 11. Employing rich, LLM-generated descriptions yields only a marginal difference in average test accuracy ($\leq 0.13\%$). This demonstrates that our method is highly robust to prompt initialization. Because our framework primarily relies on the alternating global-local prototype alignment to dynamically adapt visual representations from client data, the final model performance is driven by the FL process itself, rather than being strictly dependent on meticulously crafted initial text prompts.

*Table 12.* **Test accuracy (%) on various datasets with different backbones under the** $\rho = 0.5$ **setting.** We compare **FedSPA** against state-of-the-art baselines. **Bold** indicates the best performance within each group (Generalization or Personalization). Gray rows indicate our proposed method.

| Backbone | Method | Flower | DTD | Pets | Cars | UCF | Caltech | Food | SUN | Aircraft | EuroSAT | Mean |
|---|---|---|---|---|---|---|---|---|---|---|---|---|
| | CLIP-RN50 | 65.94 | 42.20 | 85.80 | 55.66 | 61.56 | 85.80 | 77.30 | 58.53 | 17.19 | 37.59 | 58.76 |
| | *Generalization* | | | | | | | | | | | |
| | FedPGA | 75.19 | 54.43 | 87.35 | 62.69 | 68.73 | **90.99** | **78.36** | 66.69 | 22.23 | 56.26 | 66.29 |
| | CacheFL | 68.05 | 48.29 | 87.08 | 58.02 | 68.46 | 90.98 | 78.12 | 64.15 | 19.38 | 52.27 | 63.48 |
| ResNet-50 | **FedSPA-G** | **90.01** | **59.52** | **88.01** | **63.97** | **72.61** | 90.39 | 74.05 | **66.78** | **22.65** | **69.01** | **69.70** |
| | *Personalization* | | | | | | | | | | | |
| | Tip-Adapter | 77.46 | 51.77 | 87.17 | 66.62 | 71.11 | 85.22 | 70.18 | 63.28 | 29.66 | 62.52 | 66.49 |
| | ECALP | 44.61 | 37.51 | 43.02 | 37.22 | 42.34 | 48.70 | 40.20 | 39.09 | 25.00 | 36.49 | 39.41 |
| | PromptFL | 75.07 | 54.08 | 87.83 | 60.14 | 66.92 | 90.14 | 76.85 | 64.62 | 21.04 | 59.25 | 65.59 |
| | pFedMoAP | 76.94 | 55.73 | 87.41 | 59.06 | 66.96 | 89.53 | 70.90 | 60.03 | 29.15 | 67.73 | 66.34 |
| | **FedSPA-P** | **92.43** | **66.24** | **89.08** | **71.02** | **77.02** | **92.73** | **79.39** | **72.50** | **31.58** | **70.83** | **74.28** |
| | CLIP-ViTB/16 | 71.38 | 44.39 | 89.07 | 65.27 | 66.75 | 92.94 | 86.11 | 62.55 | 24.84 | 47.77 | 65.11 |
| | *Generalization* | | | | | | | | | | | |
| | FedPGA | 79.05 | 57.92 | 91.33 | 71.48 | 76.18 | **95.29** | **86.59** | 72.54 | 30.99 | 67.27 | 72.86 |
| | CacheFL | 73.65 | 50.89 | 90.62 | 67.31 | 76.24 | 95.25 | 86.44 | 69.22 | 27.45 | 66.25 | 70.33 |
| | **FedSPA-G** | **93.71** | **63.95** | **92.15** | **72.99** | **79.33** | 93.87 | 85.00 | 71.65 | **32.52** | **77.43** | **76.26** |
| ViT-B/16 | *Personalization* | | | | | | | | | | | |
| | Tip-Adapter | 82.62 | 53.97 | 91.93 | 74.47 | 77.44 | 87.78 | 79.17 | 66.39 | 37.55 | 68.46 | 71.97 |
| | ECALP | 46.76 | 39.35 | 41.24 | 45.22 | 44.52 | 49.34 | 43.98 | 42.27 | 32.16 | 36.23 | 42.10 |
| | PromptFL | 86.80 | 58.71 | 92.54 | 69.84 | 77.07 | 94.51 | 86.02 | 70.01 | 30.13 | 66.13 | 73.17 |
| | pFedMoAP | 86.52 | 63.52 | 90.49 | 66.86 | 75.56 | 94.41 | 81.70 | 65.61 | 37.49 | 75.13 | 73.72 |
| | pFedMMA | 74.77 | 54.73 | 91.06 | 66.12 | 73.41 | 94.48 | 86.83 | 68.19 | 26.03 | 77.60 | 71.32 |
| | **FedSPA-P** | **94.61** | **70.64** | **92.84** | **78.07** | **82.39** | **96.26** | **87.02** | **77.25** | **42.52** | **78.86** | **80.04** |

# F. Performance Comparison under Moderate Heterogeneity

In the main text, we focus on the extreme non-IID setting ($\rho = 0.1$) to test the robustness of FedSPA. To provide a comprehensive evaluation across different degrees of data heterogeneity, we extend our experiments to a moderate non-IID setting with $\rho = 0.5$. All other configurations, including the 8-shot per class setting and backbone architectures (ResNet-50 and ViT-B/16), remain consistent with the main experiments. The detailed results are presented in Table 12. Compared to the extreme heterogeneity scenario ($\rho = 0.1$), the reduced distributional skew leads to an overall performance improvement across most methods, a trend that is particularly evident in the generalization metrics. However, even with improved baselines, FedSPA continues to maintain a significant lead in both generalization and personalization performance.

**Generalization Performance:** With the ResNet-50 backbone, FedSPA-G achieves a mean accuracy of $69.70\%$, outperforming the strongest baseline, FedPGA ($66.29\%$), by a margin of $3.41\%$. This advantage is even more pronounced with the ViT-B/16 backbone, where FedSPA-G reaches $76.26\%$, surpassing FedPGA ($72.86\%$). It is important to note that FedAvg-based methods, such as FedPGA and CacheFL, exhibit substantial performance gains compared to the $\rho = 0.1$ setting. This improvement stems from the fact that under moderate heterogeneity, the local visual prototypes are less skewed; consequently, the naive aggregation mechanism (FedAvg) can construct a more reliable global representation. Despite this, FedSPA-G demonstrates remarkable stability and superiority, which validates that our semantic alignment strategy captures more robust representations than simple parameter averaging, even when the data distribution becomes more favorable.

**Personalization Performance:** In terms of personalization, FedSPA-P consistently dominates all baselines. Under the ResNet-50 backbone, it achieves a mean accuracy of $74.28\%$, significantly exceeding the runner-up Tip-Adapter ($66.49\%$) by nearly $8\%$. Similarly, with ViT-B/16, FedSPA-P attains $80.04\%$, outperforming pFedMoAP ($73.72\%$) and Tip-Adapter ($71.97\%$). We also observe a similar trend among baselines relying on global aggregation, such as PromptFL. Benefiting from the higher quality of the aggregated global model under $\rho = 0.5$, PromptFL achieves significant improvements and

*Table 13.* **Detailed ablation study on semantic alignment.** We report the test accuracy (%) on each of the 10 datasets. The table compares the proposed **FedSPA** (using aligned semantic prototypes) against the **Local-Only** (w/o global guidance) and **FedAvg** (naive aggregation) variants for both generalization and personalization tasks. **Bold** indicates the best performance within each group. Gray rows highlight our proposed method.

| Method | Flower | DTD | Pets | Cars | UCF | Caltech | Food | SUN | Aircraft | EuroSAT | Mean |
|---|---|---|---|---|---|---|---|---|---|---|---|
| CLIP-RN50 | 65.94 | 42.20 | 85.80 | 55.66 | 61.56 | 85.80 | 77.30 | 58.53 | 17.19 | 37.59 | 58.76 |
| *Generalization* | | | | | | | | | | | |
| FedAvg-G | 87.17 | 53.55 | 59.01 | 50.17 | 61.27 | 83.85 | 57.83 | 56.09 | 20.19 | 62.69 | 59.18 |
| **FedSPA-G** | **89.81** | **57.98** | **87.90** | **63.35** | **72.35** | **90.47** | **77.41** | **66.96** | **21.78** | **68.85** | **69.68** |
| *Personalization* | | | | | | | | | | | |
| FedAvg-P | 91.00 | 61.48 | 63.73 | 55.10 | 66.78 | 87.57 | 60.13 | 60.80 | 33.68 | 65.32 | 64.55 |
| Local-Only | 75.23 | 62.74 | 87.82 | 69.89 | 79.29 | 94.26 | **79.86** | 75.49 | **44.12** | 59.58 | 72.82 |
| **FedSPA-P** | **93.85** | **70.88** | **89.37** | **74.89** | **81.99** | **95.01** | 79.67 | **76.71** | 41.45 | **70.97** | **77.47** |

even surpasses pFedMoAP on specific datasets (e.g., OxfordPets and StanfordCars with ViT-B/16). Consequently, FedSPA-P proves effective in leveraging global semantic prototypes for local adaptation, ensuring robust performance across varying degrees of heterogeneity.

These results, combined with the findings in the main text, confirm that FedSPA is a robust solution capable of handling a wide spectrum of non-IID scenarios, from extreme to moderate heterogeneity.

# G. Detailed Results for Main Experiments

## G.1. Detailed Ablation on Semantic Alignment Strategy

In the main text, we demonstrate the superiority of our semantic alignment module over naive aggregation (FedAvg) and Local-Only adaptation based on averaged metrics. In this section, we present detailed dataset-specific results in Table 13 to further evaluate the role of global guidance under heterogeneous settings.

**Analysis of Generalization Capabilities.** As shown in the "Generalization" section of Table 13, the limitations of naive parameter averaging are evident across specific datasets. On datasets such as OxfordPets and Food101, FedAvg-G drastically degrades performance compared to the zero-shot CLIP baseline (e.g., dropping from $85.80\%$ to $59.01\%$ on OxfordPets). This confirms that averaging divergent local prototypes destroys the pre-trained semantic structure. In contrast, FedSPA-G consistently restores and enhances performance, achieving $87.90\%$ on OxfordPets, verifying that our semantic alignment mechanism effectively aggregates global knowledge without compromising the semantic structure.

**Analysis of Personalization Capabilities.** The "Personalization" results further highlight the risk of negative transfer in federated few-shot learning. The FedAvg-P baseline ($64.55\%$ mean) significantly underperforms the Local-Only lower bound ($72.82\%$) on almost all datasets, indicating that naively aggregated global priors act as interference rather than aid for local clients. Conversely, FedSPA-P successfully bridges the gap between global generalization and local personalization. It surpasses the Local-Only baseline on 8 out of 10 datasets, with remarkable gains on EuroSAT ($+11.39\%$) and DTD ($+8.14\%$). While the Local-Only approach remains competitive on specific datasets like FGVCAircraft, FedSPA-P achieves the highest average accuracy of $77.47\%$, demonstrating its robustness as a universal solution for heterogeneous client adaptation.

## G.2. Detailed Sensitivity Analysis of Regularization Strength

In the main text, we discuss the trade-off governed by the regularization coefficient $\lambda$ in (10) based on averaged performance metrics. In this section, we present a dataset-specific analysis to substantiate our choice of the optimal regularization range. We evaluate the model performance across all ten datasets by varying $\lambda$ within the set $\{0, 0.1, 0.2, 0.5, 1.0, 2.0, 5.0\}$ under the 8-shot setting with fixed heterogeneity ($\rho = 0.1$). The detailed results, as illustrated in Figure 4, highlight the critical importance of semantic alignment. The absence of regularization ($\lambda = 0$) leads to severe performance degradation in specific datasets due to semantic drift. This is most prominent on FGVCAircraft, where accuracy drops to $24.35\%$ compared to over $45\%$ with optimal regularization, confirming that unconstrained adaptation risks catastrophic forgetting

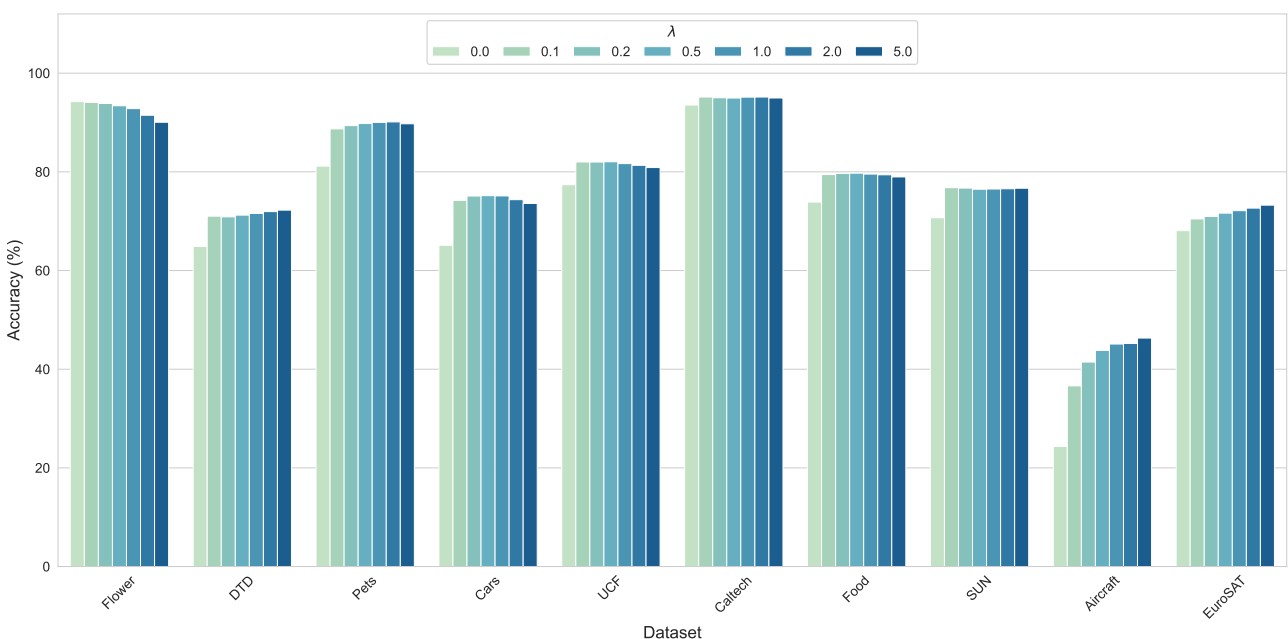

*Figure 4.* **Sensitivity analysis of regularization strength** $\lambda$ **across ten datasets.** We present a grouped bar chart illustrating dataset-specific test accuracy variations under varying regularization strengths ($\lambda$), fixed at the 8-shot setting with $\rho = 0.1$.

of the pre-trained semantic representations. As $\lambda$ increases, the performance improves significantly. However, excessive regularization ($\lambda = 5.0$) produces mixed outcomes; while it benefits datasets relying heavily on pre-trained priors (e.g., EuroSAT reaches a peak of 73.25%), it hinders necessary adaptation in others. Notably, Flowers102 shows a gradual decline from 94.23% ($\lambda = 0$) to 90.05% ($\lambda = 5.0$), suggesting that overly rigid constraints may limit the model's ability to capture fine-grained, task-specific features. Consequently, maintaining $\lambda$ within the interval $[0.5, 2.0]$ offers the most robust balance between preserving global semantic priors and local adaptability, with the average accuracy peaking at 77.96% when $\lambda = 1.0$.

### G.3. Detailed Results under Varying Shots

Complementing the averaged results presented in the main text, Figure 5 details the complete experimental results for the performance comparison under varying shots per class across all datasets. Consistent with the averaged metrics, FedSPA demonstrates remarkable robustness in low-data regimes (e.g., 2-shot and 4-shot). While it is slightly outperformed by Tip-Adapter on specific datasets such as OxfordPets, Food101, and FGVCAircraft, it remains highly competitive. In higher-shot settings, FedSPA-P maintains its superiority. With the exception of Food101, it consistently outperforms the leading baseline, pFedMoAP, across all other benchmarks. This confirms that the proposed server-side semantic alignment effectively compensates for local sample scarcity, regardless of specific dataset characteristics.

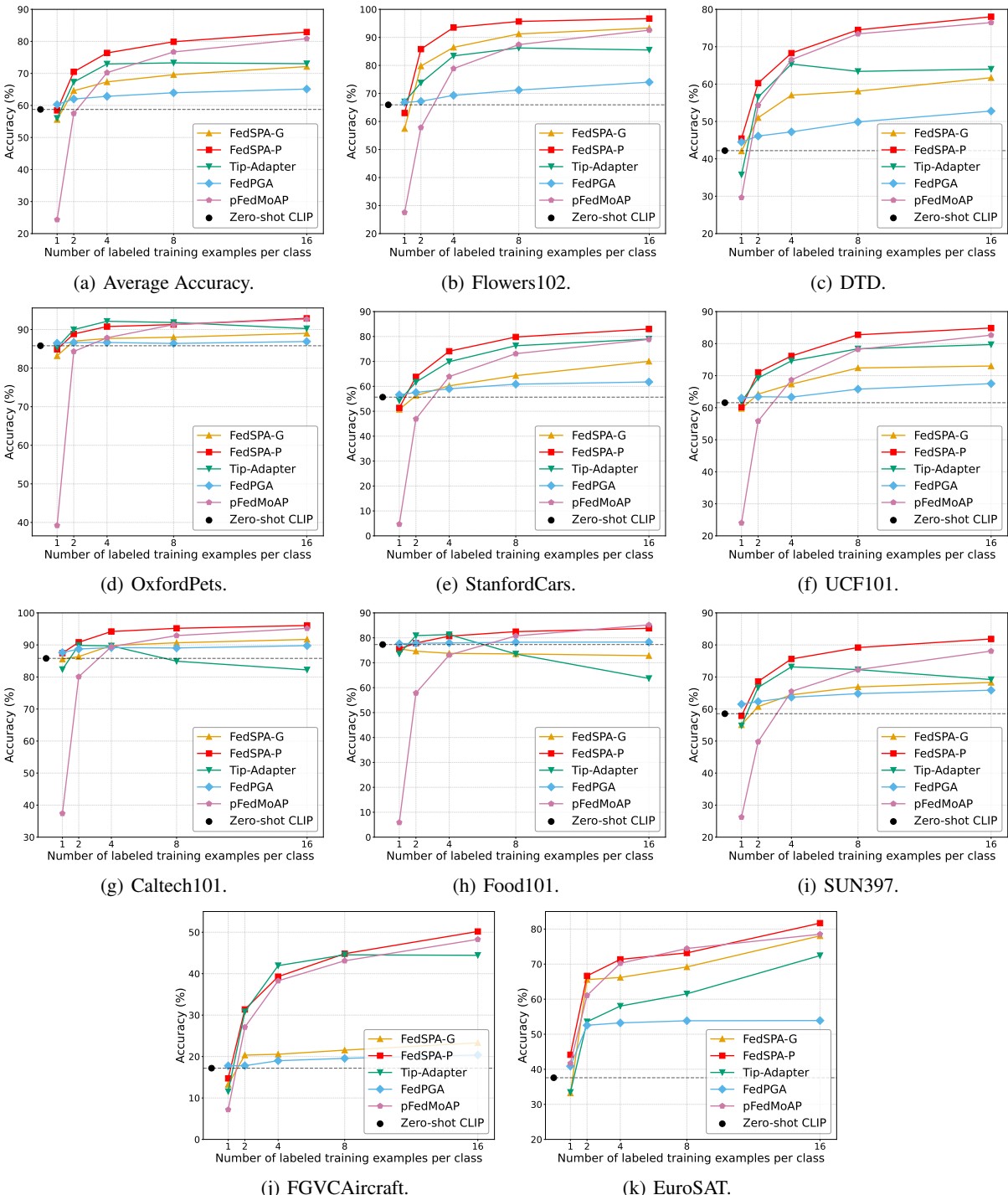

*Figure 5.* **Performance comparison under varying shots per class.** We report the average test accuracy along with the performance on each of the 10 datasets.

