# OpenReview forum: "Beyond Description: Federated Adaptation via Semantic-Visual Prototype Alignment"
_ICML.cc/2026/Conference — ICML 2026 regular_

### Official Review · Reviewer_g2Kb · 2026-03-08

**Soundness:** 3
**Presentation:** 3
**Significance:** 3
**Originality:** 2
**Overall Recommendation:** 4
**Confidence:** 4

**Summary:**

This research addresses the adaptation of Vision-Language Models (VLMs) in Federated Learning (FL). The authors propose FedSPA (Federated Adaptation via Semantic-Visual Prototype Alignment), which maintains lightweight personalized visual prototypes on the client side and optimizes global semantic prototypes via a contrastive objective (InfoNCE) on the server side. This effectively shifts the paradigm from "learning-to-describe" to "learning-to-align," mitigating client drift caused by data heterogeneity. Extensive experiments across 10 datasets demonstrate that FedSPA outperforms state-of-the-art methods in both personalization and generalization while significantly reducing computational overhead.

**Compliance With Llm Reviewing Policy:**

Affirmed.

**Key Questions For Authors:**

1. **Efficiency Benchmarking against Cache-based Methods**: In Section 4.4, Table 3 compares efficiency against FedAvg, pFedMMA, and pFedMoAP. Why were current cache-based methods like FedPGA or CacheFL  excluded from this execution time comparison?
2. **Privacy Implications of Visual Prototypes**: The framework requires uploading class-wise local centroids (visual prototypes) to the server. Have you evaluated their vulnerability to feature inversion or membership inference attacks?
3. **Communication Overhead Analysis**: While client-side computation is shown to be efficient, the paper lacks specific metrics on communication volume. Could you provide the total parameter size and traffic (in MB) required to reach convergence compared to PromptFL?

**Limitations:**

yes

**Strengths And Weaknesses:**

Strengths：
1. Innovative Paradigm Shift: The transition from optimizing static prompts to dynamic alignment of global semantics with heterogeneous visual distributions is highly effective.
2. Superior Efficiency and Performance: Extensive experiments across 10 datasets demonstrate that FedSPA outperforms state-of-the-art methods in both personalization and generalization. Meanwhile, It achieves a 44.4x speedup over FedAvg.

Weaknesses：
1. **Limited Technical Novelty**: The proposed framework appears to be an incremental combination of established concepts—such as prototype-based Federated Learning, CLIP adaptation via prototypes, and contrastive representation alignment—rather than a fundamentally new methodology. Consequently, the contribution feels more like an application of existing techniques to the VLM-FL domain than a significant algorithmic breakthrough.

---

> ### Author Rebuttal · Authors · 2026-03-29
>
> **[Q1] Limited Technical Novelty.**
>
> **[A1]** FedSPA shifts the VLM-FL paradigm from computationally heavy prompt tuning ("learning-to-describe") to efficient prototype alignment ("learning-to-align"), resolving the performance-efficiency bottleneck. Its core novelty lies in two inherently coupled designs tailored for VLMs:
>
> 1. **VLM-Tailored Federated Architecture Design:** Unlike adapter/prompt-based methods requiring expensive deep-layer backpropagation, FedSPA treats the VLM as a frozen black box. It optimizes only low-dimensional visual prototypes on the client side and semantic prototypes on the server side.
> 2. **Alternating Global-Local Prototype Alignment:** Traditional cache/prototype-based FL struggles with data heterogeneity. We introduce alternating global-local prototype alignment to replace the traditional weighted aggregation method and achieve SOTA performance in both personalization and generalization
>
> To further validate this, we introduce the Intra-Class Cross-Client Cosine Distance to measure prototype divergence across clients. Under extreme heterogeneity ($\rho=0.1$), naive local-only updates cause severe divergence (distance: 0.1862). In contrast, **FedSPA significantly reduces this distance to 0.1254 (a 32.7% reduction)**. This confirms that our method successfully pulls divergent local spaces into a unified consensus, while simultaneously capturing distinct personalized decision boundaries as demonstrated in Table 1.
>
> **[Q2] Efficiency Benchmarking against Cache-based Methods.**
>
> **[A2]** We excluded cache-based methods from Table 3 because FedSPA matches their extreme efficiency. Table 3 is designed to demonstrate computational speedups specifically against adapter/prompt-based methods burdened by expensive deep-layer backpropagation. Since cache-based methods are similarly lightweight, comparing speeds is redundant. Instead, as shown in Table 1, FedSPA’s true advantage over Cache-based Methods lies in its exceptional robustness under data heterogeneity, achieving up to +7.44% higher accuracy than CacheFL (ResNet-50).
>
> **[Q3] Privacy Implications of Visual Prototypes.**
>
> **[A3]** Transmitting class-wise visual prototypes inherently offers robust privacy defenses compared to raw data or gradients:
>
> 1. **Deep Embedding Abstraction:** The uploaded prototypes are highly abstracted deep embeddings extracted from the terminal layers of a frozen VLM (e.g., CLIP ViT-B/16). Unlike raw pixels or shallow features, these deep representations deliberately discard fine-grained spatial information, making it exceptionally difficult for attackers to execute model inversion or reconstruct raw images [R1], [R2].
> 2. **Instance-Level Anonymization:** Averaging visual features across multiple images within a class washes out instance-specific granular details (e.g., backgrounds, unique identities), making it highly resistant to data reconstruction attacks. The transmission of prototypes is a widely adopted practice in prototype-based FL frameworks [R3], [R4].
> 3. **Architectural Privacy Superiority:** Information exchange is fundamental to any FL system. However, compared to traditional FL (transmitting gradients) and Split FL (transmitting intermediate activations), our method ensures higher privacy by communicating only highly abstracted, class-wise prototypes.
>
> **[Q4] Communication Overhead Analysis.**
>
> **[A4]** We calculated the exact communication overhead (upload + download) for 10 global rounds per client using CLIP ViT-B/16 ($d=512$, single-precision floats):
>
> *   **PromptFL:** Transmits soft prompts ($16 \times 512$ params). Traffic is constant at **0.63 MB** across all datasets.
> *   **FedSPA (Ours):** Transmits class-wise prototypes ($N \times 512$ params). Traffic dynamically scales with class count $N$.
>
> **Total Traffic per Client (10 Rounds):**
>
> *   EuroSAT ($N=10$): **0.39 MB**
> *   Aircraft / Food101 ($N=100/101$): **~3.95 MB**
> *   SUN397 ($N=397$): **15.51 MB**
>
> While FedSPA incurs higher overhead on fine-grained datasets with numerous categories, the absolute maximum traffic for the **entire training process** is only ~15.5 MB. This is extremely lightweight and negligible under modern bandwidths compared to transmitting full FL models (>300 MB). Given the significant improvements in both model accuracy and local training efficiency (Table 3), this communication trade-off is highly justifiable.
>
>
>
> > [R1] NoPeek: Information leakage reduction to share activations in distributed deep learning. ICDMW 2020.
> >
> > [R2] ResSFL: A Resistance Transfer Framework for Defending Model Inversion Attack in Split Federated Learning. CVPR 2022.
> >
> > [R3] Federated Prototype Guided Adaption for  Vision-Language Models. ICASSP 2025.
> >
> > [R4] Personalized Federated Learning with Feature Alignment and Classifier Collaboration. NeurIPS 2023

---

> > ### Author Rebuttal · Reviewer_g2Kb · 2026-04-03
> >
> > My question has been resolved, but based on the author's response and the opinions of other reviewers, I have decided to maintain the current score.

---

> > > ### Author Response · Authors · 2026-04-04
> > >
> > > We sincerely thank you for your time and appreciate your constructive feedback, which has helped us improve our work. While we understand your decision to maintain the score due to other reviewers' feedback, please note that in the second rebuttal phase, we have submitted point-by-point additional experiments and analyses aimed at resolving those remaining issues.
> > >
> > > Most notably, to address the privacy concerns, we upgrade FedSPA with a **dual-layer privacy defense**. We formalize FedSPA with **$(\epsilon, \delta)$-Local Differential Privacy (LDP)** via feature-level Gaussian noise injection, providing theoretical bounds for instance-level privacy. Crucially, by averaging features into prototypes before transmission, FedSPA irreversibly masks individual data, providing a natural defense against reconstruction attacks. Our empirical evaluation across all 10 datasets shows that FedSPA maintains high utility, with only a marginal ~1.09% accuracy drop under small LDP noise settings. For the detailed analysis, **please refer to our response to Reviewer gwUb [A1].**
> > >
> > > Furthermore, a brief summary of the newly added experiments and clarifications in response to other reviewers is as follows:
> > >
> > > *   **Global Generalization vs. Prompt-tuning:** We evaluate FedSPA against baselines like PromptFL and pFedMoAP. Results show that under extreme heterogeneity ($\rho=0.1$), FedSPA effectively prevents local overfitting and achieves SOTA global generalization (reaching 69.59% accuracy vs. pFedMoAP's 34.60%).
> > > *   **Efficiency Comparison:** We add a comparison with cache-based baselines (CacheFL, FedPGA). The results demonstrate that FedSPA achieves the same magnitude of computational acceleration (~3.60s execution time) while yielding significantly higher accuracy (+5.6% to +7.4%).
> > > *   **Extended Convergence Analysis:** We extend the federated training from 10 to 15 global rounds. The results empirically prove that FedSPA safely and rapidly converges within 10 rounds and exhibits robust stability without oscillation in subsequent rounds.
> > > *   **Clarification on Novelty (vs. FedRep):** We clarify our fundamental distinction from prior local-global methods, emphasizing our paradigm shift from Network-layer level decomposition (which requires deep back-propagation) to Representation-space level decomposition (a strictly training-free backbone).
> > >
> > > For the complete experimental results of these summaries, **please kindly refer to our detailed response to Reviewer iVdB.** Thank you once again for your effort in reviewing our paper!

---

### Official Review · Reviewer_gwUb · 2026-03-12

**Soundness:** 3
**Presentation:** 3
**Significance:** 3
**Originality:** 3
**Overall Recommendation:** 4
**Confidence:** 3

**Summary:**

To address the issues of high computational cost and ineffective knowledge aggregation, the paper proposes the FedSPA method. To tackle computational overhead, the paper freezes the pre-trained model’s backbone and only trains the visual prototypes and semantic prototypes. The visual prototypes are initialized from the features of images processed by the image encoder across individual clients, while the semantic prototypes are initialized from the output of the text encoder using the template "a photo of a [cls]". To solve the ineffective knowledge aggregation problem, the paper replaces simple parameter averaging with contrastive learning-based semantic alignment (InfoNCE) and incorporates a regularization term, enabling the server to aggregate a robust, and semantically consistent global representation from highly heterogeneous client knowledge.

**Compliance With Llm Reviewing Policy:**

Affirmed.

**Final Justification:**

The concerns are partially resolved, as I explained in the acknowledgement section.

**Key Questions For Authors:**

1.	$L_k$ is defined as the empirical risk on client k, but the document does not elaborate on its specific form here. Could you clarify the mathematical expression for $L_k$?
2.	In your experimental setup, you used "a photo of a [cls]" as the basis for initializing $z_0$. Theoretically, other prompt engineering functions could also be used to initialize $z_0$. Has this been attempted?

**Limitations:**

In the proposed method of the paper, two N×d-dimensional matrices are maintained. The claim of low computational overhead is based on scenarios with a relatively small number of classes N. However, when the scale of the classification task increases,the computational overhead may similarly rise significantly, potentially affecting communication and computational efficiency.

**Strengths And Weaknesses:**

Strength

1.The framework appears practical and effectively reduces computational cost.During the training process, the backbone remains frozen. And during inference, the model does not require repeated computation of text inputs through the text encoder.

2.The paper replaces simple parameter averaging with contrastive learning-based semantic alignment, which gives the server-side process a clear and meaningful purpose.

Weakness

1.The application of prototypes in federated settings may introduce risks of information leakage for the model.

2.Moving the computation of the text encoder to the server alleviates the burden on clients, but the class names are also revealed simultaneously, which may indicate shortcomings in the model's security.

3.The article has only two baselines regarding cache-based methods.It should provide more baselines to demonstrate the model's generalizability.

4.The reproducibility of the method currently presents certain limitations. Although the algorithm flow is provided, the lack of publicly available code and specific settings for some key hyperparameters makes it challenging to independently reproduce the experimental results reported in the paper.

---

> ### Author Rebuttal · Authors · 2026-03-29
>
> **[Q1] Prototypes may introduce privacy risks and information leakage.**
>
> **[A1]** Transmitting class-wise prototypes offers robust privacy via **Deep Embedding Abstraction** and **Instance-Level Anonymization**. Due to space constraints, please refer to our detailed privacy analysis in **[A3] to Reviewer g2Kb**.
>
> **[Q2] Server-side text encoding reveals raw class names, posing security risks.**
>
> **[A2]** This security risk can be entirely avoided in our framework. Since the text encoder is only used once during the "Global Initialization" phase, clients can locally compute these initial semantic vectors $\mathbf{z}_n^{(0)}$ using their local class names and upload only the resulting vectors. The server then aggregates these vectors to initialize $\mathbf{Z}^{(0)}$. Thus, the server never accesses raw text, completely eliminating label leakage risks.
>
> **[Q3] The paper only compares against two cache-based baselines; more are needed to demonstrate generalizability.**
>
> **[A3]** To the best of our knowledge, FedPGA (2025) and CacheFL (2025) are currently the only existing SOTA baselines specifically designed for this setting. If the reviewer is aware of other relevant baselines, we would be grateful for the references and will gladly include them.
>
> **[Q4] Reproducibility is limited due to the lack of publicly available code and specific hyperparameter settings.**
>
> **[A4]** For hyperparameters, please refer to **Appendix A (Tables 4 and 5)**, where we have extensively documented all configurations (e.g., hardware, data partition $\rho$, optimizers, learning rates, communication rounds, and method-specific parameters $\alpha, \beta, \lambda$). Regarding the code, we firmly commit to open-sourcing the complete codebase once the paper is made public.
>
> **[Q5] The mathematical expression for the empirical risk $\mathcal{L}_k$ on client $k$.**
>
> **[A5]** Since our objective is $N$-way image classification, $\mathcal{L}_k$ is the standard Cross-Entropy loss computed over the client's local dataset $\mathcal{D}_k$, utilizing the personalized prediction scores $\mathbf{S}_k$ from Eq. (3). The explicit expression is:
>
> $ \mathcal{L}_k(\mathbf{V}_k, \mathbf{Z}) = \frac{1}{|\mathcal{D}_k|} \sum_{(\mathbf{x}, y) \in \mathcal{D}_k}\ell_{CE} \Big( \mathbf{S}_k(f(\mathbf{x}); \mathbf{V}_k, \mathbf{Z}), y \Big) $
>
> **[Q6] Has the framework been tested with other prompt engineering functions for initialization?**
>
> **[A6]** We replace the simple template with rich, LLM-generated (GPT) visual descriptions for each class to initialize $\mathbf{z}^{(0)}_n$ [R1] [R2]. As shown in the table below, this yields only a marginal accuracy difference ($\leq 0.13\%$). This demonstrates that FedSPA is highly **robust to prompt initialization**. Because our framework deeply relies on alternating global-local prototype alignment, the final performance is driven by the federated learning process itself rather than meticulously crafted initial text prompts.
>
> | Method       | Prompt Initialization  | Average Acc. (%)  |
> | :----------- | :--------------------- | :---------------: |
> | **FedSPA-G** | Simple / LLM-generated | 69.59 / **69.72** |
> | **FedSPA-P** | Simple / LLM-generated | 79.89 / **79.92** |
>
> **[Q7] Maintaining two $N \times d$-dimensional matrices may cause computational and communication overhead to rise significantly when the number of classes $N$ increases.**
>
> **[A7]** While FedSPA's overhead scales with $N$, empirical evaluations confirm it remains highly efficient even under extreme scaling (e.g., SUN397 with $N=397$). As shown in the table below, the maximum total communication budget (upload + download for 10 rounds) is only **15.51 MB**, which is negligible under modern bandwidths compared to transmitting full FL models (>300 MB). By trading this marginal communication increase, FedSPA achieves a massive **>100x computational speedup** over pFedMoAP (4.17s vs. 441.40s on an RTX 4080). This efficiency makes FedSPA suitable for real-world edge devices with **strictly constrained computing resources**.
>
> | Dataset (Classes $N$) | Method                | Total Traffic (MB) | Execution Time (s) |
> | :-------------------- | :-------------------- | :----------------: | :----------------: |
> | **EuroSAT** ($N=10$)  | pFedMoAP / **FedSPA** |  0.63 / **0.39**   |  17.59 / **0.21**  |
> | **Food101** ($N=101$) | pFedMoAP / **FedSPA** |  0.63 / **3.95**   |  40.06 / **1.21**  |
> | **SUN397** ($N=397$)  | pFedMoAP / **FedSPA** |  0.63 / **15.51**  | 441.40 / **4.17**  |
>
>
>
> > [R1] Dual Memory Networks:  A Versatile Adaptation Approach for Vision-Language Models. CVPR2024.
> >
> > [R2] Efficient and Context-Aware Label Propagation for Zero-/Few-Shot Training-Free Adaptation of Vision-Language Model. ICLR2025.

---

> > ### Author Rebuttal · Reviewer_gwUb · 2026-04-03
> >
> > I would like to thank the authors for addressing the concerns related to Security, Reproducibility, Robustness to Initialization, and Scalability. Their detailed responses and additional empirical results have significantly clarified the technical mechanisms and practical effectiveness of FedSPA.
> > However,it should be noted that Deep Embedding Abstraction and Instance-Level Anonymization do not fundamentally solve the privacy issue, such defenses increase the attacker's computational cost rather than providing a formal privacy guarantee at the mathematical level. Thus, I will maintain my score.

---

> > > ### Author Response · Authors · 2026-04-04
> > >
> > > **[Q1] Empirical defenses like Abstraction and Anonymization only increase the attacker's computational costs and lack formal mathematical privacy guarantees.**
> > >
> > > **[A1]** We sincerely appreciate the reviewer's insightful suggestion. To provide formal privacy guarantees, we equip FedSPA with **Local Differential Privacy (LDP)** via the Feature-level Gaussian Mechanism. Crucially, FedSPA provides a **dual-layer privacy protection**: a theoretical LDP bound via noise injection, and a strong empirical defense via prototype aggregation.
> > >
> > > 1. **Theoretical Guarantee (Feature-level LDP):**
> > >
> > > To protect instance-level privacy, we inject calibrated Gaussian noise into each individual visual feature prior to class-wise prototype aggregation. Since the frozen VLM features $f(\mathbf{x})$ are L2-normalized ($||f(\mathbf{x})||_2 = 1$), the global sensitivity for replacing any single image is strictly bounded by $\Delta = 2$. Therefore, clients apply the following mechanism to each feature:
> > > $$ \tilde{f}(\mathbf{x}) = f(\mathbf{x}) + \mathcal{N}(0, \sigma^2 \mathbf{I}) $$
> > > This mechanism mathematically guarantees $(\epsilon, \delta)$-Local Differential Privacy ($\delta=10^{-5}$). Because DP is immune to post-processing, the subsequent operations inherently inherit this theoretical LDP bound.
> > >
> > > 2. **Empirical Privacy Enhancement (Prototype Aggregation):**
> > >
> > > Furthermore, we emphasize that the practical privacy protection in FedSPA is significantly stronger than what the theoretical worst-case $\epsilon$ suggests. The theoretical LDP bound is calculated based on the extreme worst-case scenario (i.e., a class containing only $1$ sample). However, in our design, the server primarily receives aggregated class-wise **prototypes** rather than isolated individual features. Except for the rare extreme case where a client possesses exactly one sample for a specific class, these prototypes are weighted averages of multiple local samples (up to 8). This aggregation acts as a powerful empirical privacy safeguard. From an attacker's perspective, feature inversion on an aggregated, noisy prototype is an inherently ill-posed problem, making it nearly impossible to reconstruct any specific raw image.
> > >
> > > 3. **Empirical Evaluation & Robustness:**
> > >
> > > While achieving single-digit $\epsilon$ in LDP for high-dimensional representations without destroying utility remains a known open challenge, we evaluated FedSPA across all 10 datasets under varying noise multipliers ($\sigma$) and their corresponding theoretical worst-case bounds ($\epsilon \approx 9.68/\sigma$):
> > >
> > > | Noise Regime        | Noise ($\sigma$) | Worst-case $\epsilon$ | Acc (%) w/ $T=5$ | Acc (%) w/ $T=10$ | Acc (%) w/ $T=20$ |
> > > | :------------------ | :--------------- | :-------------------- | :--------------- | :---------------- | ----------------- |
> > > | Baseline (No Noise) | $0$              | $\infty$              | 79.89            | -                 | -                 |
> > > | Small Noise         | $0.10$           | $\approx 96.8$        | 78.80            | 78.99             | 79.00             |
> > > | Medium Noise        | $0.24$           | $\approx 40.3$        | 76.92            | 77.35             | 77.73             |
> > > | Large Noise         | $0.48$           | $\approx 20.2$        | 72.96            | 73.86             | 74.58             |
> > >
> > > As demonstrated, FedSPA shows robustness to DP noise. Under the original setting ($T=5$), it experiences only a marginal 1.09% accuracy drop under small noise. Furthermore, by appropriately increasing the local optimization epochs (e.g., from $T=5$ to $T=10$ or $T=20$)—which is computationally highly feasible given FedSPA's training-free backbone—we can further enhance the model's robustness. The underlying mechanism is that increased local optimization steps allow the personalized visual prototypes to better converge, effectively smoothing out the zero-mean Gaussian noise. This enables FedSPA to sustain a relatively high accuracy (74.58%) even under the large noise regime.
> > >
> > > 4. **Privacy Advantages over Existing FL Paradigms:**
> > >
> > > Compared to existing FL paradigms, FedSPA demonstrates superior robustness against privacy leakage. Specifically:
> > > (1) Traditional FL methods require transmitting high-dimensional model weights or gradients, which are vulnerable to severe privacy leakage via gradient inversion attacks.
> > > (2) Prompt-tuning-based FL methods (e.g., PromptFL and pFedMoAP) require sharing textual prompts, which inevitably exposes sensitive class names and label semantics to the server, whereas FedSPA can avoid this by computing text features strictly locally.
> > > (3) Compared to other prototype-based FL approaches (e.g., FedPGA), FedSPA introduces no additional privacy risks. Furthermore, FedSPA can effectively safeguard local data privacy through feature-level Local Differential Privacy (LDP) and the inherent aggregation properties of the prototypes.
> > >
> > > We will explicitly include the privacy analysis and the corresponding ablation study in the Appendix.

---

### Official Review · Reviewer_3nQA · 2026-03-12

**Soundness:** 3
**Presentation:** 3
**Significance:** 2
**Originality:** 3
**Overall Recommendation:** 4
**Confidence:** 4

**Summary:**

This paper addresses key challenges in Federated Learning (FL), including the efficient adaptation of vision-language models in federated learning settings under data heterogeneity. To overcome these limitations, the authors propose Federated Adaptation via Semantic-Visual Prototype Alignment (FedSPA). The approach restricts optimization solely to a lightweight visual prototypes on the client side and aligns global semantic prototypes with heterogeneous visual distributions on the server side to construct a robust global classifier. Experimental results on a suite of 10 diverse image classification datasets demonstrate that the proposed model achieves state-of-the art performance.

**Compliance With Llm Reviewing Policy:**

Affirmed.

**Final Justification:**

The authors have addressed my concerns, and the reported results are SOTA. Therefore, I decide to raise my score.

**Key Questions For Authors:**

1. Why was the InfoNCE loss chosen for server-side alignment over simpler aggregation methods like weighted averaging?
2. How sensitive is the contrastive learning process to the initialization of the global prototypes?
3. How does the framework handle extreme label heterogeneity where a client may have zero samples for certain classes?

**Limitations:**

The problem definition could be further refined to emphasize the significance of adaptation of VLMs in federated learning settings. While the introduction addresses the key challenges, a more explicit connection to practical applications would strengthen the motivation behind the work.

**Strengths And Weaknesses:**

Strengths:
1.	Freezing the VLM backbone and optimizing only lightweight prototypes obtain the best performance than the comparative related methods.
2.	Extensive experiments are conducted to evaluate the effectiveness of the proposed model.

Weakness:
1. The paper motivates its study by referring to broad and well-known challenges in federated learning, namely efficient adaptation and data heterogeneity, but it does not clearly articulate a specific and well-defined technical gap that existing methods fail to address.
2. Limited methodological novelty: The core idea of decoupling local and global representations has been extensively explored in federated learning. The adaptation of this concept to VLMs using visual prototypes and contrastive alignment on the server is an engineering choice, not a scientific innovation.
3. The framework reduces client-side learning by updating a single linear layer of prototypes. The author should provide a theoretical or empirical justification for why this highly constrained update is sufficient to capture the complex distribution shifts present in highly heterogeneous client data.
4. There is no visualization or analysis of the learned prototype spaces. It remains unclear whether client prototypes for the same class indeed converge, and whether the server’s alignment mechanism actually reduces client representation divergence.
5. The server only ever sees client prototypes, and the server-side alignment module performs contrastive learning directly with these visual prototypes. This method still suffer from a significant distribution shift. The author should explain how to guarantee the global prototypes can effectively guide personalization on the client rather than leading to a collapse of global knowledge.
6. The total number of communication rounds is very low.
7. The evaluation on larger-scale datasets or more complex tasks remains unverified.

---

> ### Author Rebuttal · Authors · 2026-03-29
>
> **[Q1] Unclear specific technical gap beyond broad FL challenges.**
>
> **[A1]** The specific gap is the strict trade-off between efficiency and performance in VLM-FL. Deep-tuning methods (Prompt/Adapter) handle heterogeneity well but are computationally prohibitive for edge clients. Cache-based methods are extremely fast but highly sensitive to data heterogeneity. FedSPA bridges this by freezing the client backbone (updating only lightweight visual prototypes) and introducing a novel server-side semantic alignment module to replace the traditional weighted aggregation method.
>
> **[Q2] Decoupling local/global representations seems like an engineering choice.**
>
> **[A2]** Far from a mere engineering trick, our decoupling is an algorithmic design fundamentally grounded in the VLM architecture. While existing VLM-FL methods (e.g., FedPGP, pFedMoAP) restrict decoupling to the prompt space, FedSPA achieves a prototype-level decoupling tailored for the VLM. Specifically, we uniquely decouple representations across the global textual semantic space and the local image visual space. By alternating the alignment of these dual spaces, FedSPA naturally achieves SOTA in both personalization and generalization.
>
> **[Q3] Why is updating a single prototype layer sufficient for highly heterogeneous data?**
>
> **[A3]** CLIP’s pre-trained embedding space is highly linearly separable. In heterogeneous regimes, deep back-propagation often severely distorts this manifold [R1]. Updating a single linear layer acts as robust linear probing, discovering optimal local decision boundaries without destroying the pre-trained feature space. Few-shot adaptation works (e.g., Tip-Adapter, TDA [R2]) empirically validate that single-layer updates on frozen CLIP features perfectly capture complex shifts.
>
> **[Q4] Lack of analysis confirming client prototype convergence or reduced divergence.**
>
> **[A4]** We introduce the **Intra-Class Cross-Client Cosine Distance**. Under extreme heterogeneity ($\rho=0.1$), naive local-only updates cause severe divergence. In contrast, FedSPA reduces the average cross-client distance by **32.7%**, confirming that our mechanism successfully pulls divergent local spaces into a unified consensus.
>
> | Method            | Average Distance ($\downarrow$) |
> | :---------------- | :-----------------------------: |
> | **Local-Only**    |             0.1862              |
> | **FedSPA (Ours)** |           **0.1254**            |
> | *(Reduction)*     |           ***32.7%***           |
>
> **[Q5] Risk of global knowledge collapse from contrastive alignment with skewed prototypes.**
>
> **[A5]** There is no such risk in FedSPA. We explicitly prevent global knowledge collapse via the stability regularizer $\mathcal{L}_{reg}$ (Eq. 9, 10). Acting as a "semantic anchor", it restricts representations from drifting into skewed local spaces. Detailed ablations in Fig. 3 and 4 demonstrate its effectiveness.
>
> **[Q6] Only 10 communication rounds evaluated.**
>
> **[A6]** FedSPA converges to SOTA in just 10 rounds, proving its extreme communication efficiency. Furthermore, 10 rounds matches the standard configurations in baseline works (PromptFL, pFedMoAP).
>
> **[Q7] Unverified on larger-scale datasets or complex tasks.**
>
> **[A7]** Our 10-dataset benchmark already covers massive classes (SUN397: 397 categories), fine-grained tasks (Aircraft, Flower, Pets), and severe domain shifts (EuroSAT, DTD) under extreme non-IID settings ($\rho=0.1$).
>
> **[Q8] Why use InfoNCE instead of simpler weighted averaging?**
>
> **[A8]** Simple weighted averaging fails under extreme data heterogeneity because client prototypes drift into divergent, disjoint spaces. We have explicitly shown this limitation in Table 2, where naive "FedAvg-G" drops to 59.18% compared to FedSPA-G's 69.68%. InfoNCE acts as an active structural aligner. It uses client-uploaded visual prototypes as references to dynamically refine global semantic prototypes.
>
> **[Q9] Sensitivity to global prototype initialization.**
>
> **[A9]** The framework is highly robust to prompt initialization. Please refer to our detailed ablation in **[A6] to Reviewer gwUb** (shows a negligible $\le 0.13\%$ accuracy gap).
>
> **[Q10] Handling missing classes in extreme label heterogeneity.**
>
> **[A10]** For a missing class $X$, the client simply skips local visual updates but still downloads the aligned global semantic prototype from the server. This enables robust zero-shot inference for missing classes  (Eq. 3).
>
> **[Q11] Lack of explicit connection to practical applications.**
>
> **[A11]** In the revision, we will explicitly ground the Introduction in high-impact applications, such as personalized visual assistants on edge devices (e.g., smartphones), which demand extreme computational efficiency and adaptation to highly idiosyncratic data.
>
>
>
> > [R1] Fine-Tuning can Distort Pretrained Features and Underperform Out-of-Distribution. ICLR 2022
> >
> > [R2] Efficient Test-Time Adaptation of Vision-Language Models. CVPR2024

---

> > ### Author Rebuttal · Reviewer_3nQA · 2026-04-03
> >
> > I have read the authors’ rebuttal and the authors have addressed all of my concerns.

---

> > > ### Author Response · Authors · 2026-04-04
> > >
> > > Thank you so much for your supportive message and for confirming that your concerns have been fully resolved. We truly appreciate the time you dedicated to reviewing our paper and your recognition of our work.

---

### Official Review · Reviewer_iVdB · 2026-03-13

**Soundness:** 2
**Presentation:** 3
**Significance:** 2
**Originality:** 2
**Overall Recommendation:** 4
**Confidence:** 4

**Summary:**

This paper proposes FedSPA, a federated adaptation framework for vision–language models based on semantic–visual prototype alignment. The method restricts client-side optimization to lightweight visual prototypes while maintaining global semantic prototypes on the server. The server aligns semantic prototypes with client visual prototypes through a contrastive objective to handle heterogeneous data distributions. Experiments on multiple image classification datasets show improvements over several federated prompt learning baselines and report lower computational cost.

**Compliance With Llm Reviewing Policy:**

Affirmed.

**Final Justification:**

Thank you for the detailed follow-up. The additional experiments address several of my earlier concerns. While I still have some reservations about novelty and privacy, I believe the paper is now sufficiently supported. I encourage the authors to include these clarifications and results in the revised paper, and I update my rating to Weak Accept.

**Key Questions For Authors:**

1) The server runs 100 gradient epochs per round on the proxy dataset. What is the server-side wall-clock time per round, and how does it scale with the number of clients K and classes N?

2) Can the authors evaluate FedSPA-G against prompt tuning baselines (PromptFL, pFedMoAP) to provide a complete generalization comparison?

3)  Clients upload class-wise visual feature means every round. How does this compare to raw data exposure, and what privacy guarantees can the authors provide for this upload?

**Limitations:**

No.
The Impact Statement is a generic placeholder and no limitations section exists.

**Strengths And Weaknesses:**

**Strengths**

1) The paper studies federated adaptation for vision–language models, which is an important problem given the increasing interest in parameter-efficient VLM adaptation in federated settings.

2) Restricting local optimization to lightweight visual prototypes is a practical design that reduces client-side computation compared to methods that require back-propagation through large backbones.

3) The empirical evaluation includes comparisons with several federated prompt learning and personalized federated learning baselines.

**Weaknesses**

1) The number of server-side optimization epochs is unablated. The server runs 100 gradient epochs per round to update the global semantic prototypes, yet the paper provides no sensitivity analysis on this hyperparameter. It is unclear whether this setting is necessary or whether fewer epochs would achieve comparable performance, and how this choice scales with the number of clients K and classes N.

2) Clients upload class-wise mean visual features each round. While these statistics are more compact than raw data, feature-level representations may still reveal distributional information about local datasets. The paper does not discuss this potential privacy exposure or compare it against methods that avoid transmitting feature representations.

3) Novelty relative to prior local-global decomposition methods is unclear. Maintaining global shared and local personalized components is well-established in FedPGP[1], FedOTP[2], and FedRep[3]. The paper does not clearly articulate what FedSPA contributes beyond replacing aggregation with contrastive server-side alignment.


[1] Cui et al. Harmonizing Generalization and Personalization in Federated Prompt Learning. ICML 2024.

[2] Li et al. Global and Local Prompts Cooperation via Optimal Transport for Federated Learning. CVPR 2024.

[3] Collins et al. Exploiting Shared Representations for Personalized Federated Learning. ICML 2021.

---

> ### Author Rebuttal · Authors · 2026-03-29
>
> **[Q1] Server-side optimization epochs ($T_{server}=100$) ablation & wall-clock time scaling w.r.t $T_{server}, K, N$?**
>
> **[A1]** We defaulted to $T_{server}=100$ based on the standard FL assumption that servers possess abundant compute. Crucially, optimizing the server-side InfoNCE loss strictly involves lightweight, parallelized matrix multiplications over low-dimensional semantic prototypes ($K \times N \times d$, $d=512$). The empirical time complexity scales linearly as $\mathcal{O}(T_{server} \cdot K \cdot N \cdot d)$. To demonstrate this, we measure the per-round wall-clock time and accuracy across varying $T_{server}$, $K$, and $N$ on an RTX 4080.
>
> > **Metrics below are formatted as (Time (s) / FedSPA-G Acc (%) / FedSPA-P Acc (%) ):**
>
> | Dataset (N, K)    | T = 10                   | T = 50                    | **T = 100 (Default)**     |
> | ----------------- | ------------------------ | :------------------------ | :------------------------ |
> | **Pets (37,10)**  | ( 0.99 / 87.63 / 90.63 ) | ( 4.61 / 87.87 / 91.04 )  | ( 9.29 / 88.01 / 91.29 )  |
> | **UCF (101,10)**  | ( 2.92 / 70.34 / 82.09 ) | ( 10.30 / 71.27 / 82.11 ) | ( 30.42 / 72.40 / 82.76 ) |
> | **Cars (196,20)** | ( 9.95 / 64.95 / 78.32 ) | ( 49.54 / 63.87 / 78.74 ) | ( 99.28 / 63.10 / 78.55 ) |
>
> The execution time scales smoothly and linearly. Remarkably, model accuracy remains highly robust, showing negligible fluctuations even with drastically reduced $T_{server}$. In resource-constrained server scenarios, one can safely use $T_{server}=10$ or $50$ to ensure extreme computational efficiency without sacrificing performance.
>
> **[Q2] Do class-wise visual features leak distributional info compared to raw data? What privacy guarantees exist?**
>
> **[A2]** Transmitting class-wise prototypes offers robust privacy via **Deep Embedding Abstraction** and **Instance-Level Anonymization**. Due to space constraints, please refer to our detailed privacy analysis in **[A3] to Reviewer g2Kb**.
>
> **[Q3] Global-local decomposition is established (e.g., FedPGP, FedOTP, FedRep). What is FedSPA's contribution beyond contrastive alignment?**
>
> **[A3]** While global-local decomposition has historical precedents, FedSPA introduces a fundamentally novel paradigm tailored for VLMs:
>
> 1. **Modal-Level Decoupling:** Unlike FedRep which decomposes unimodal DNN parameters (e.g., base layers vs. heads), FedSPA uniquely decouples representations across the textual semantic space (global) and the image visual space (local).
> 2. **Algorithmic Shift from Prompts to Prototypes:** Methods like FedPGP/FedOTP rely on prompt tuning, which inevitably requires expensive gradient back-propagation through the VLM backbone. FedSPA shifts to prototype alignment, restricting client computation exclusively to low-dimensional matrices. This yields a >40$\times$ speedup (Table 3).
> 3. **Dual Capability:** Our alternating alignment achieves SOTA performance in both personalization and generalization—a dual capability that existing prompt-based frameworks struggle to maintain under extreme data heterogeneity.
>
> **[Q4] Evaluate FedSPA-G generalization against prompt tuning baselines (PromptFL, pFedMoAP).**
>
> **[A4]** PromptFL and pFedMoAP are fundamentally designed for **Personalized Federated Learning (PFL)**. Under extreme data heterogeneity ($\rho=0.1$), their learnable soft prompts strictly overfit to skewed local client distributions. Consequently, evaluating their aggregated prompts on global text data yields sub-optimal generalization, making direct comparisons unfair.
> This highlights FedSPA’s unique architectural advantage: local personalization is handled by both personalized visual prototypes $\mathbf{V}_k$ and global semantic prototypes $\mathbf{Z}$ (Eq. 3), while global generalization is driven independently by global semantic prototypes $\mathbf{Z}$ (Eq. 2). As Table 1 shows, both FedSPA-P and FedSPA-G achieve SOTA performance.
>
> **[Q5] Provide a specific Impact Statement and add a Limitations section.**
>
> **[A5]** We apologize for this oversight. We will add the following to the final manuscript:
>
> *   **Limitations:** FedSPA requires semantically meaningful labels to initialize global semantic prototypes via the text encoder (Eq. 1). In domains where labels are merely anonymized IDs (e.g., class_1) without textual descriptions, the framework cannot fully exploit the VLM's language priors. Furthermore, server computation scales linearly with class and client sizes ($N, K$). Developing efficient contrastive alignment via class/client sampling for ultra-large-scale datasets is left for future work.
> *   **Broader Impact:** FedSPA significantly lowers computational/communication barriers for democratizing VLMs in federated edge networks. By eliminating deep-backbone back-propagation on edge clients, it contributes to "Green AI" by drastically reducing carbon footprints. Ensuring ethical use and robust privacy measures remains a shared responsibility in decentralized ML.

---

> > ### Author Rebuttal · Reviewer_iVdB · 2026-04-03
> >
> > Thank you for the rebuttal. I appreciate the added clarifications. The discussion of server-side optimization and limitations is helpful, but my core concerns remain only partially addressed. In particular, the privacy argument for uploading class-wise visual prototypes remains largely qualitative and does not provide formal guarantees or empirical leakage analysis. The novelty claim is clearer after the rebuttal, but I still do not find the distinction from prior local-global decomposition methods fully convincing. In addition, the requested generalization comparison against prompt-tuning baselines is still missing. I also agree with other reviewers that the limited evaluation over only 10 communication rounds and the lack of a direct efficiency comparison against other lightweight cache-based baselines remain relevant concerns. Therefore, my overall evaluation remains unchanged.

---

> > > ### Author Response · Authors · 2026-04-04
> > >
> > > We sincerely thank the reviewer for the rigorous feedback. To fully address your remaining concerns, we have carefully conducted point-by-point additional experiments corresponding to each of your requests.
> > >
> > > **[Q1] Privacy argument lacks empirical leakage analysis or formal guarantees.**
> > >
> > > **[A1]** We address this by incorporating a **dual-layer privacy protection** framework. First, we formalize FedSPA with **$(\epsilon, \delta)$-Local Differential Privacy (LDP)** by injecting calibrated Gaussian noise into individual $L_2$-normalized visual features. Second, our class-wise prototype aggregation serves as a robust empirical privacy defense, inherently preventing the server from observing single raw features. Evaluations across all 10 datasets demonstrate that FedSPA preserves high utility, experiencing only a ~1.09% accuracy drop under small noise regimes. Furthermore, our lightweight training-free design allows clients to cost-effectively increase local epochs (e.g., to $T=10$) to smooth out DP noise, further boosting robustness under large noise regimes. (Due to character limits, please refer to our detailed privacy analysis in **Response to Reviewer gwUb [A1]**).
> > >
> > > **[Q2] Missing generalization comparison against prompt-tuning baselines.**
> > >
> > > **[A2]** We evaluate the global generalization capability of PromptFL and pFedMoAP on the global test set under extreme data heterogeneity ($\rho=0.1$). The average accuracy across all 10 datasets is presented below:
> > >
> > > | Method              | PromptFL | pFedMoAP | FedSPA-G  |
> > > | :------------------ | :------: | :------: | :-------: |
> > > | **Global Acc. (%)** |  64.05   |  34.60   | **69.59** |
> > >
> > > As shown, prompt-tuning methods struggle significantly with global generalization. While PromptFL achieves moderate accuracy via prompt aggregation, pFedMoAP—a framework strictly designed for personalized FL—severely overfits to skewed local client distributions, resulting in drastic performance degradation on the global test set. In contrast, FedSPA effectively avoids this issue by leveraging a decoupled global semantic prototype ($\mathbf{Z}$) optimized via contrastive alignment, achieving SOTA generalization while simultaneously preserving personalization capability.
> > >
> > > **[Q3] Distinction from prior local-global decomposition (e.g., FedRep) is not fully convincing.**
> > >
> > > **[A3]** We clarify that FedSPA is fundamentally distinct from prior local-global methods through a paradigm shift from **Network-layer level** to **Representation-space level** decomposition:
> > >
> > > 1.  Prior Methods (e.g., FedRep): Perform Network-layer decomposition. They partition the model into a shared deep backbone and local MLPs. Crucially, they still require computationally expensive gradient back-propagation through deep neural layers.
> > > 2.  FedSPA: Performs Cross-modal Representation-space decomposition. We decouple the modalities at the embedding level: visual features remain local ($\mathbf{V}_k$), while text semantics are globally aligned ($\mathbf{Z}$).
> > > 3.  **The fundamental difference:** FedSPA enforces a strictly "Training-free Backbone". Our client-side optimization requires zero back-propagation through the VLM. It replaces deep gradient updates with highly lightweight matrix multiplications on low-dimensional prototypes.
> > >
> > > **[Q4] Limited evaluation over only 10 communication rounds.**
> > >
> > > **[A4]** We extend the federated training from 10 to 15 global communication rounds under extreme heterogeneity ($\rho=0.1$). The average accuracy across the 10 datasets is shown below:
> > >
> > > | Global Round          |  10   |  11   |  12   |  13   |  14   |  15   |
> > > | :-------------------- | :---: | :---: | :---: | :---: | :---: | :---: |
> > > | **FedSPA-G Acc. (%)** | 69.59 | 69.62 | 69.57 | 69.29 | 69.55 | 69.56 |
> > > | **FedSPA-P Acc. (%)** | 79.89 | 79.90 | 80.00 | 79.98 | 80.13 | 80.03 |
> > >
> > > The results clearly indicate that FedSPA safely converges within 10 rounds, exhibiting stability without oscillation in subsequent rounds. We emphasize that $I=10$ communication rounds is a standard evaluation protocol widely adopted by recent SOTA few-shot FL frameworks (e.g., PromptFL, pFedMoAP, pFedMMA).
> > >
> > > **[Q5] Lack of efficiency comparison against cache-based baselines.**
> > >
> > > **[A5]** We add an efficiency comparison against cache-based baselines.
> > >
> > > | Method       | Execution Time (s) | Avg. Acc (%) $\uparrow$ |
> > > | :----------- | :----------------: | :---------------------: |
> > > | CacheFL      |        3.80        |          62.15          |
> > > | FedPGA       |        3.86        |          63.96          |
> > > | **FedSPA-G** |      **3.60**      |        **69.59**        |
> > >
> > > As demonstrated, FedSPA achieves the same magnitude of computational acceleration as leading cache-based methods. However, FedSPA's true advantage lies in its robustness under extreme non-IID settings. By effectively aligning semantic and visual spaces, FedSPA significantly outperforms cache-based baselines in model accuracy (+5.6% to +7.4%) without sacrificing computational efficiency.

---

### Decision · Program_Chairs · 2026-04-30

**Decision:**

Accept (regular)

**Comment:**

The paper receives four reviews with 4x weak accept ratings. The reviewers generally appreciate the practical federated design as well as strong empirical performance. However, the reviewers also point out issues related to novelty and privacy. Specifically, the reviewers view the approach as an incremental extension from prior local-global decomposition methods. The authors argue that the approach is a paradigm shift from network-layer level to representation-space level decomposition. In terms of privacy leakage, the reviewers are concerned that uploading class-wise visual prototypes may leak distributional information but the authors argue that sharing prototypes instead of raw data has reduced leakage risk. The rebuttal has mostly addressed the reviewers' questions, and all reviewers vote for weak acceptance. Considering that the reviews are mostly positive and the findings of this work are useful to the federated learning community, the AC recommends acceptance based on the conditions that the authors include the discussions related to differences with prior methods and privacy leakage risk in the final version.